# Few-Shot Class-Incremental Learning via Training-Free Prototype Calibration

**Qi-Wei Wang, Da-Wei Zhou, Yi-Kai Zhang, De-Chuan Zhan, Han-Jia Ye**[*]
State Key Laboratory for Novel Software Technology
Nanjing University, Nanjing, 210023, China
{wangqiwei,zhoudw,zhangyk,zhandc,yehj}@lamda.nju.edu.cn

## Abstract

Real-world scenarios are usually accompanied by continuously appearing classes with scarce labeled samples, which require the machine learning model to incrementally learn new classes and maintain the knowledge of base classes. In this Few-Shot Class-Incremental Learning (FSCIL) scenario, existing methods either introduce extra learnable components or rely on a frozen feature extractor to mitigate catastrophic forgetting and overfitting problems. However, we find a tendency for existing methods to misclassify the samples of new classes into base classes, which leads to the poor performance of new classes. In other words, the strong discriminability of base classes distracts the classification of new classes. To figure out this intriguing phenomenon, we observe that although the feature extractor is only trained on base classes, it can surprisingly represent the *semantic similarity* between the base and *unseen* new classes. Building upon these analyses, we propose a *simple yet effective* Training-frEE calibratioN (TEEN) strategy to enhance the discriminability of new classes by fusing the new prototypes (*i.e.*, mean features of a class) with weighted base prototypes. In addition to standard benchmarks in FSCIL, TEEN demonstrates remarkable performance and consistent improvements over baseline methods in the few-shot learning scenario. Code is available at: https://github.com/wangkiw/TEEN

## 1 Introduction

Deep Neural Networks (*i.e.*, DNNs) have achieved impressive success in various applications [16, 19, 36, 55, 54, 5], but they usually rely heavily on *static and pre-collected large-scale* datasets (*e.g.*, ImageNet [11]) to achieve this success. However, the data in real-world scenarios usually arrive continuously. For example, the face recognition system is required to authenticate existing users, and meanwhile, new users are continually added [30]. The scene where the model is required to continually learn new knowledge and maintain the ability on old tasks is referred to as *Class-Incremental Learning (CIL)* [41]. The main challenge of CIL is the notorious *catastrophic forgetting problem* [14], where the model forgets old knowledge as it learns new ones.

Many methods have been designed to overcome *catastrophic forgetting* in CIL from different perspectives, *e.g.*, knowledge distillation [35], parameter regularization [22, 53], and network expansion [50, 47, 46]. These methods require new tasks to contain *sufficient* labeled data for supervised training. However, collecting enough labeled data in some scenarios is challenging, making the conventional CIL methods hard to deploy [30]. For example, the face recognition system can only collect very few facial images of a new user due to privacy reasons. Therefore, a more realistic and practical incremental learning paradigm, *Few-Shot Class-Incremental Learning (FSCIL)* [42], is

---

[*]Han-Jia Ye is the corresponding author.

37th Conference on Neural Information Processing Systems (NeurIPS 2023).

proposed to address the problem of class-incremental learning with limited labeled data. Figure 1a gives a detailed illustration of FSCIL.

In addition to the *catastrophic forgetting problem* [14], FSCIL will suffer the *overfitting problem* because the model can easily overfit the very few labeled data of new tasks. Previous works [57, 58, 61] have adopted prototype-based methods [40, 33] to conquer the limited data problem. These methods freeze the feature extractor trained on base classes when dealing with new classes and use the prototypes of new classes as the corresponding classifier weights. The frozen feature extractor can alleviate the *catastrophic forgetting problem* and the prototype classifier can circumvent the *overfitting problem*. However, the inherent problem of learning with few-shot data is challenging to depict precisely the semantic information of a new category with limited data, making prototypes of new classes inevitably biased [27]. An intuitive reason for the biased prototypes is that the feature extractor has not been optimized for the new classes. For example, the feature extractor trained on "cat" and "dog" can not precisely depict the feature of a new class "bird" especially when the instances of "bird" are incredibly scarce [2]. Many prototype adjustment methods [27, 57, 58, 15, 61, 1, 24] are dedicated to rectifying the biased representations of the new classes. These methods usually focus on designing complex pre-training algorithms to enhance the compatibility of representations [58, 59, 10], or complicated trainable modules that better adapt the representation of the new classes [57, 15, 61, 1], all of which rely on *significant training costs* in exchange for improved model performance.

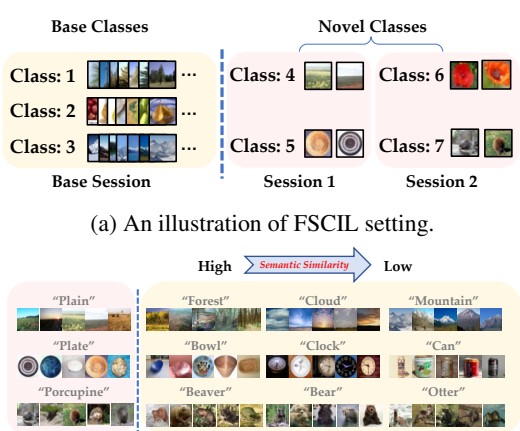

(a) An illustration of FSCIL setting.

(b) An illustration of the *semantic similarity*.

Figure 1: (a) The base session contains *sufficient* samples of base classes for training. The incremental sessions (*i.e.*, sessions after the base session) only contain *few-shot* samples of new classes. FSCIL aims to obtain a unified classifier over all seen classes. Notably, the data from previous sessions are unavailable. (b) For example, considering the new classes "Plain", "Plate" and "Porcupine" in CIFAR100 [23], the corresponding most similar base classes selected by the similarity can depict the really semantic similarity. The three most similar base classes are selected by computing the cosine similarity between base prototypes and novel prototypes. The results show *the feature extractor only trained on the base classes can also well represent the semantic similarity between the base and new classes.*

However, we find that although existing methods perform well on the widely used performance measure (*i.e.*, the average accuracy across all classes), *they usually exhibit poor performance on new classes*, which suggests that the calibration ability of existing methods could be improved. In other words, existing methods neglect the performance of new classes. A direct reason for this negligence is the current *unified* performance measure is easily overwhelmed by the dominated base classes (*e.g.*, 60 base classes in the CIFAR100 dataset). However, the more recent tasks are usually more crucial in some real-world applications. For example, the new users in the recommendation system are usually more important and need more attention [45]. The *important yet neglected* new classes' performance inspires us to pay more attention to it.

To better understand the performance of current FSCIL methods [42, 6, 9, 58, 57], we explicitly evaluate and analyze the low performance of existing methods on *new classes* and empirically demonstrate *the instances of new classes are prone to be predicted into base classes*. To further understand this intriguing phenomenon, we revisit the representation of the feature extractor and find that the feature extractor trained only on base classes can already represent the semantic similarity between the base and new classes well. As shown in Figure 1b, although the novel classes are unavailable during the training of the feature extractor, the *semantic similarity* between the base and novel classes can also be well-represented. However, existing methods have been obsessed with designing complex learning modules and disregard this off-the-shelf *semantic similarity*.

---

[2] "new class" and "novel class" in this paper mean the classes emerging after the base session.

Based on the above analysis, we propose to leverage the overlooked semantic similarity to explicitly enhance the discriminability of new classes. Specifically, we propose a *simple yet effective* Training-frEE prototype calibratioN (*i.e.*, TEEN) strategy for biased prototypes of new classes by fusing the biased prototypes with weighted base prototypes, where the weights for the base prototypes are *class-specific and semantic-aware*. Notably, TEEN does not rely on any extra optimization procedure or model parameters once the feature extractor is trained on the base classes with a vanilla supervised optimization objective. Besides, the excellent calibration ability and training-free property make it a plug-and-play module.

Our contributions can be summarized as follows:

- We empirically find that the lower performance of new classes is due to misclassifying the samples of new classes into corresponding similar base classes, which is intriguing and missing in existing studies.

- We propose *a simple yet effective training-free* calibration strategy for new prototypes, which not only achieved a higher average accuracy but also improved *the accuracy of new classes* (**10.02% $\sim$ 18.40% better than the runner-up FSCIL method**).

- We validate TEEN on benchmark datasets under the standard FSCIL scenario. Besides, TEEN shows competitive performance under the few-shot learning scenario. The consistent improvements demonstrate TEEN's remarkable calibration ability.

## 2   Related Works

### 2.1   Class Incremental Learning

The model in the Class-Incremental Learning (*i.e.*, CIL) scenario is required to learn new classes without forgetting old ones. Save representative instances in old tasks (*i.e.*, exemplars) and replay them in new tasks is a simple and effective way to maintain the model's discriminability ability on old tasks [20, 2, 37]. Furthermore, knowledge distillation [17, 35, 25, 49] is widely used to maintain the knowledge of the old model by enforcing the output logits between the old model and the new one to be consistent. iCaRL [35] uses the knowledge distillation as a regularization item and replays the exemplars when learning the feature representation. Many methods follow this line and design more elaborate strategies to replay exemplars [29] and distill knowledge [21, 12]. Recently, model expansion [28, 50, 47, 46, 39, 60] have been confirmed to be effective in CIL. The most representative method [50] saves a single backbone and freezes it for each incremental task. The frozen backbone substantially alleviates the catastrophic forgetting problem.

### 2.2   Few-Shot Learning

The model in Few-Shot Learning (*i.e.*, FSL) [48] scenario is required to learn new classes with limited labeled data. Existing methods usually achieve this goal either from the perspective of optimization [13, 31, 3] or metric learning [40, 43, 26, 27, 56]. The core thought of optimization-based methods is to equip the model with the ability to fast adaptation with limited data. The metric-based methods focus on learning a unified and general distance measure to depict the semantic similarity between instances. Besides, [51] proposes to use Gaussian distribution to model each feature dimension of a specific class and sample augmented features from the calibrated distribution. Based on these augmented features, [51] train a logistic regression and achieve competitive performance. However, TEEN can outperform [51] in most settings and without any training cost when recognizing new classes.

### 2.3   Few-Shot Class-Incremental Learning

The model in Few-Shot Class-Incremental Learning (*i.e.*, FSCIL) [42] scenario is required to incrementally learn new knowledge with limited labeled data. Many existing methods are dedicated to designing learning modules to train a more powerful feature extractor [58] or adapting the representation of the instances of new classes [57, 9, 61, 1]. Besides, TOPIC [42] utilizes a neural gas network to alleviate the challenging problems in FSCIL. [8] adopts word embeddings as semantic information and introduces a distillation-based FSCIL method. IDLVQ [6] proposes to utilize quantized reference vectors to compress the old knowledge and improve the performance in FSCIL. However, all these

methods overlook the abundant semantic information in base classes and the poor performance in new classes. In this study, we aim to take a small step toward filling this gap. The most related work to TEEN is [1]. However, it relies on optimizing a regularization-based objective function to implicitly utilize the semantic information. As opposed to [1], TEEN takes advantage of the empirical observation and gets rid of the optimization procedure and is thus more efficient and effective.

## 3 Preliminaries

### 3.1 Definition and notations

In FSCIL [42], we assume there exists $T$ sessions in total, including a base session (*i.e.*, the first session) and $T - 1$ incremental sessions (*i.e.*, sessions after the first session). We denote the training data in the base session as $\mathcal{D}_0$ and the training data in the incremental sessions as $\{\mathcal{D}_1, \mathcal{D}_2, \ldots, \mathcal{D}_{T-1}\}$. For the training data $\mathcal{D}_i$ in the $i$-th session, we further notate it with $\{(x_j, y_j)\}_{j=1}^{N_i}$ and corresponding label space with $\mathcal{C}_i$. Note that only training data $\mathcal{D}_i$ is available in the $i$-th session. Accordingly, the testing data and testing label space in session $i$ can be denoted as $\mathcal{D}_i^{test}$ and $\mathcal{C}_i^{test}$. To better evaluate the model's discriminability on all seen tasks, the testing label space $\mathcal{C}_i^{test}$ of $i$-th session contains all seen classes during training, *i.e.*, $\mathcal{C}_i^{test} = \bigcup_{j=0}^{i} \mathcal{C}_j$. An incremental session can also be denoted as a $N$-way $K$-shot classification task, *i.e.*, $N$ classes and $K$ labeled examples for each class. Note that the training label spaces between different sessions are disjoint, *i.e.*, for any $i, j \in [0, T-1]$ and $i \neq j, \mathcal{C}_i \bigcap \mathcal{C}_j = \varnothing$.

Compared to conventional CIL, FSCIL only requires the model to learn new classes with *limited* labeled data. On the other hand, compared to conventional Few-Shot Learning (FSL), FSCIL requires the model to continually learn the knowledge of new classes while *retaining the knowledge of previously seen classes*. We introduce the related works on CIL, FSL and FSCIL in supplementary due to the space limitation.

The model in FSCIL can be decoupled into a feature encoder $\phi_\theta(\cdot)$ with parameters $\theta$ and a linear classifier $W$. Given a sample $x_j \in \mathbb{R}^D$, the feature of $x_j$ can be denoted as $\phi_\theta(x_j) \in \mathbb{R}^d$. For a $N$-class classification task, the output logits of a sample $x_j$ can be denoted as $\mathcal{O}_j = W^\top \phi_\theta(x_j) \in \mathbb{R}^N$ where $W \in \mathbb{R}^{d \times N}$.

### 3.2 Prototypical Network

ProtoNet [40] is a widely used method in few-shot learning problems. It computes the mean feature $c_k$ of a class $k$ (*e.g.*, class prototype) and uses the class prototype to represent the corresponding class:

$$c_k = \frac{1}{\text{Num}_k} \sum_{y_j=k} \phi_\theta(x_j) \tag{1}$$

$\text{Num}_k$ denotes the number of samples in class $k$. For a classification task with $N$ classes, the classifier can be represented by the $N$ prototypes, *i.e.*, $W = [c_1, c_2, \ldots, c_N]$. Following the [58, 57, 61, 1], we freeze the feature extractor $\phi_\theta$ trained on the base task and plug the class prototype into the classifier while dealing with a new class. The frozen feature extractor can alleviate catastrophic forgetting and the plug-in updating of the classifier can circumvent the overfitting problem relatively.

## 4 A Closer Look at FSCIL

In this session, we comprehensively analyze current FSCIL methods from a *decoupled* perspective. Although the previous updating paradigm of *extractor-frozen and prototypes-plugged* can achieve adequate average accuracy in all classes, there also exist some shortcomings in it. In this section, we empirically show that the current methods (*e.g.*, [57, 58]) are generally not effective in new classes. Furthermore, we take a step toward understanding the reason for the low performance in new classes.

### 4.1 Understanding the reason for poor performance in new classes

To better understand the performance of existing methods, we first measure the performance by average accuracy on all classes, base classes and new classes, respectively. As illustrated in Figure 2,

Table 1: Detailed prediction results of **False Negative Rate/False Positive Rate** (%) on CIFAR100 [23] dataset. The analysis results are from session 1 because new classes do not exist in session 0. Exceedingly high **FPR** and relatively low **FNR** show the instances of new classes are easily misclassified into base classes and the instances of base classes are also easily misclassified into base classes. TEEN can achieve relatively lower **FPR** than baseline methods, which demonstrates the validity of the proposed calibration strategy. Please refer to the supplementary for results on *mini*ImageNet and CUB200.

| Session | 1 | | 2 | | 3 | | 4 | | 5 | | 6 | | 7 | | 8 | |
|---|---|---|---|---|---|---|---|---|---|---|---|---|---|---|---|---|
| **FNR/FPR** | FNR | FPR | FNR | FPR | FNR | FPR | FNR | FPR | FNR | FPR | FNR | FPR | FNR | FPR | FNR | FPR |
| ProtoNet [40] | 2.13 | 67.40 | 4.42 | 67.40 | 6.68 | 60.67 | 8.28 | 58.90 | 10.25 | 56.68 | 11.70 | 54.47 | 12.57 | 51.66 | 13.78 | 51.80 |
| CEC [57] | 2.32 | 70.40 | 4.38 | 66.20 | 6.18 | 62.20 | 7.50 | 58.65 | 9.72 | 56.00 | 11.30 | 53.60 | 12.12 | 51.40 | 13.48 | 51.78 |
| FACT [58] | 2.05 | 66.60 | 3.88 | 61.70 | 5.58 | 56.80 | 7.23 | 55.05 | 8.85 | 53.64 | 9.83 | 51.13 | 10.45 | 48.83 | 11.75 | 49.27 |
| TEEN | 4.03 | **57.40** | 7.40 | **52.50** | 9.35 | **45.00** | 11.58 | **40.75** | 14.00 | **40.80** | 15.78 | **39.23** | 16.33 | **36.91** | 18.75 | **36.65** |

Table 2: Detailed prediction results of **TNR/TBR** (%) on CIFAR100 [23] dataset. The analysis results are from session 1 because new classes do not exist in session 0. For new classes, we only consider the 10 most similar base classes out of 60 base classes. For base classes, we suppose $C_i$ new classes exist in the current incremental session $i$. We only consider the most similar $\lfloor 20\% \times C_i \rfloor$ new classes. Class similarity adopts cosine similarity between different class prototypes. TEEN can achieve relatively lower **TBR** than baseline methods, which demonstrates the validity of the proposed calibration strategy. Please refer to the supplementary for results on *mini*ImageNet and CUB200.

| Session | 1 | | 2 | | 3 | | 4 | | 5 | | 6 | | 7 | | 8 | |
|---|---|---|---|---|---|---|---|---|---|---|---|---|---|---|---|---|
| **TNR/TBR** | TNR | TBR | TNR | TBR | TNR | TBR | TNR | TBR | TNR | TBR | TNR | TBR | TNR | TBR | TNR | TBR |
| ProtoNet [40] | 3.47 | 73.01 | 5.02 | 61.79 | 7.06 | 55.05 | 8.08 | 52.95 | 7.14 | 53.27 | 8.56 | 51.96 | 8.53 | 49.18 | 9.21 | 50.91 |
| CEC [57] | 2.87 | 71.70 | 4.53 | 61.08 | 5.66 | 58.26 | 6.35 | 55.16 | 6.07 | 54.12 | 6.92 | 52.53 | 8.24 | 49.96 | 8.20 | 51.64 |
| FACT [58] | 2.33 | 70.00 | 3.32 | 61.01 | 5.50 | 55.17 | 6.88 | 50.55 | 6.57 | 50.52 | 7.72 | 49.30 | 8.88 | 46.99 | 9.31 | 48.48 |
| TEEN | 2.16 | **66.77** | 2.53 | **55.14** | 3.55 | **44.87** | 3.85 | **37.68** | 3.57 | **39.47** | 4.02 | **37.07** | 4.76 | **33.83** | 4.85 | **35.04** |

our first observation is that **the average accuracy across base classes is extremely higher than the accuracy in new classes**. The inconsistent performance between the base and new classes is caused by the frozen feature extractor and biased new prototypes. The former forces the model to *overfit* base classes and the latter causes the model to *underfit* the new classes.

To determine the real cause of the low performance on new classes, we further investigate "*What base classes are the new classes incorrectly predicted into?*" and get our second observation.

Our second observation is that **the prototype-based classifier misclassifies the new classes to their corresponding most similar base classes with high probability**, *i.e.*, many instances of new classes are closer to their nearest base prototype than corresponding target prototypes

To verify this observation, we first analyze the detailed prediction results from a decoupled perspective. Specifically, we treat all base classes as a "positive class" and all new classes as a "negative class" and transform the FSCIL problem into a two-class classification task. We compute the false negative rate (*i.e.*, **FNR**) and false positive rate (*i.e.*, **FPR**) of each binary prediction task on each incremental session. The FNR and FPR in the confusion matrix is defined as follows:

$$\text{FNR} = \frac{\text{FN}}{\text{TP} + \text{FN}} \times 100\%, \ \text{FPR} = \frac{\text{FP}}{\text{FP} + \text{TN}} \times 100\% \quad (2)$$

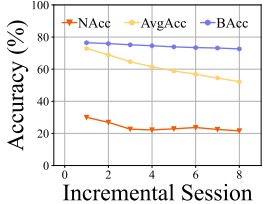

Figure 2: The performance of FACT [58] on CIFAR100 dataset. **NAcc,BAcc,AvgAcc** mean the average accuracy on *new classes, base classes, and all classes*, respectively.

As shown in Table 1, the FPR is far greater than FNR illustrating new classes are generally misclassified into base classes but the base classes are generally misclassified into base classes. On the basis of this conclusion, we further explore the details of misclassification. To better illustrate the analysis, we define "misclassified **T**o most similar **B**ase classes **R**atio" (*i.e.*, **TBR** ) for new classes and "misclassified **T**o most similar **N**ew classes **R**atio" (*i.e.*, **TNR** ) for base classes. Specifically, considering the misclassified instances in base classes and new classes respectively, the TBR and

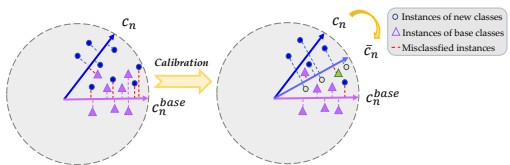

(a) An *qualitative* illustration of the calibration

(b) An *quantitative* illustration of the calibration

| Incremental Session | 1 | | | 2 | | |
|---|---|---|---|---|---|---|
| Sample type | UC | W→R | R→W | UC | W→R | R→W |
| Base Class | 93.08 | 8.51 | **76.92** | 87.51 | 7.95 | **68.03** |
| New Class | 6.92 | **91.49** | 23.08 | 12.49 | **92.05** | 31.97 |

Figure 3: (a) The $\bigcirc$ represents the samples of new classes and the $\triangle$ represents the samples of base classes. The dotted lines between samples and prototypes (*i.e.*, $c_n, c_n^{base}, \bar{c}_n$) represent the corresponding classification results. Blue and purple dotted lines represent samples that are correctly classified, and red dotted lines represent misclassification. The yellow circles (*i.e.*, **W→R** samples) and green triangles (*i.e.*, **R→W** samples) in the right figure are samples with prediction changes after calibration. (b) The detailed ratio (%) of base and new classes with regard to three types of samples(*i.e.*, **UC** samples, **W→R** samples, **R→W** samples). Only two incremental sessions (*i.e.*, session 1 and session 2) of CIFAR100 are listed here for convenience. Please refer to the supplementary for results on more incremental sessions and more datasets.

TNR are defined as follows:

$$\text{TBR} = \frac{M_b}{N_n} \times 100\%, \ \text{TNR} = \frac{M_n}{N_b} \times 100\% \tag{3}$$

$N_n$ means the number of misclassified instances of new classes. $M_b$ is the number of instances misclassified into most similar base classes in $N_n$ samples. $M_n$ and $N_b$ have similar meaning. As shown in Table 2, the TBR is higher and the TNR is relatively low, which strongly underpins our second observation. To our knowledge, we are the first to explain the reason for the low performance of new classes in existing methods and observe that the samples of new classes are easily misclassified into the most similar base classes.

To summarize, we empirically demonstrate that existing methods perform poorly on new classes and find that this is because the model tends to misclassify new class samples into the most similar base classes. Combining existing analysis (*i.e.*, *the samples of new classes tend to be classified into the most similar base classes*) and observations (*i.e.*, *the well-trained feature extractor on base classes can also well represent the semantic similarity between the base and new classes*), we propose to use the frozen feature extractor as a *bridge* and calibrate the biased prototypes of new classes by fusing the biased new prototypes with the well-calibrated base prototypes.

## 5 Similarity-based Prototype Calibration

Based on the common assumptions in FSCIL [42], sufficient instances of base classes are available during the training of the feature extractor. We argue the *sufficient* data in the base session contains abundant semantic information (*i.e.*, a *sufficient* number of classes) and the base prototypes are well-calibrated (*i.e.*, a *sufficient* number of instances for each class). The above analysis leads to a natural question:

*Can we leverage the well-calibrated prototypes in the base session for a new prototype calibration?*

As the previous methods overlook the low performance of new classes caused by the corresponding biased prototypes, we propose to *explicitly calibrate* these biased prototypes with the help of the well-calibrated base prototypes. The off-the-shelf semantic similarity serves as a bridge between the base prototypes and the new ones. In the following sections, we introduce the details of fusing the well-calibrated base prototypes and ill-calibrated new prototypes to calibrate the biased ones. Afterwards, we analyze the effects of this training-free prototype calibration on new classes and base classes, respectively.

### 5.1 Fusing the biased prototypes with calibration item

We assume there exist $B$ classes in the base session and $C$ classes in an incremental session, *e.g.*, $B = 60, C = 5$ in the CIFAR100 dataset. Without loss of generality, we only consider the base session and the first incremental session for simplification. Other incremental sessions can be obtained in the same way. Following the notations in section 3.1, the empirical prototype of $i$-th class can

be notated as $c_i$. Therefore, the base prototypes are $c_b(0 \leq b \leq B - 1)$ and the new prototypes are $c_n(B \leq n \leq B + C - 1)$. As the base session contains *sufficient* classes and *sufficient* samples for each class, the model trained on the base session can capture the distribution of base classes and obtain well-calibrated prototypes for base classes. Generally, the empirical prototype of base classes can be regarded as approximately consistent with the expected class representation. However, due to the limited data in incremental sessions, the empirical prototypes of the novel classes are considered to be severely biased.

Based on the analysis of prototypes, we only calibrate biased prototypes in incremental sessions. Given a new class prototype $c_n(B \leq n \leq B + C - 1)$, the calibrated new prototype $\bar{c}_n$ can be notated by:

$$\bar{c}_n = (1 - \alpha)c_n + \alpha \Delta c_n \tag{4}$$

The calibration item $\Delta c_n$ is a component of base prototypes. The hyperparameter $\alpha$ controls the calibration strength of biased prototypes. Smaller $\alpha$ means the calibrated prototype reflects more of the original biased prototype, while larger $\alpha$ means the calibrated prototype heavily incorporates the base prototypes. Motivated by the observations in section 4.1, the similarity between well-calibrated base prototypes and ill-calibrated new prototypes contains auxiliary side information about the new classes. Therefore, we use weighted base prototypes to represent the calibration item $\Delta c_n$ and enhance the discriminability of biased prototypes. Specifically, we compute the cosine similarity $S_{b,n}$ between a new class prototype $c_n$ and a base prototype $c_b$:

$$S_{b,n} = \frac{c_b \cdot c_n}{\|c_b\| \cdot \|c_n\|} \cdot \tau \tag{5}$$

where $\tau$ ($\tau > 0$) is the scaling hyperparameter controlling the weight distribution's sharpness. The weight of a new prototype $c_n$ with respect to a base prototype $c_b$ is the softmax results over all base prototypes:

$$w_{b,n} = \frac{e^{S_{b,n}}}{\sum_{i=0}^{B-1} e^{S_{i,n}}} \tag{6}$$

Finally, the calibration of biased prototypes of new classes can be formulated as follows:

$$\bar{c}_n = (1 - \alpha)\,c_n + \alpha\,\Delta c_n = (1 - \alpha)\,c_n + \alpha \sum_{b=1}^{B-1} w_{b,n}c_b \tag{7}$$

Notably, the above prototype rectification procedure is a *training-free* calibration strategy because it does not introduce any learning component or training parameters. Figure 3a gives an intuitive description of the calibration effect.

## 5.2 Effect of calibrated prototypes

Intuitively, $w_{b,n}$ is larger when a new prototype $c_n$ and base prototype $c_b$ are more similar. Given a new prototype $c_n$, we assume *the most similar base prototype* with $c_n$ is the base prototype $c_n^{base}$. Therefore, given a new prototype $c_n$, $\sum_{b=1}^{B-1} w_{b,n}c_b \approx c_n^{base}$ when the scaling hyperparameter $\tau$ is large enough. From the perspective of a biased prototype, a calibrated prototype $\bar{c}_n$ will be aligned to its most similar base prototypes $c_n$ with a proper $\tau$.

To further comprehend the effect of TEEN on the predictions of base and new classes respectively, we define three types of test samples according to whether the prediction results change after TEEN: with unchanged predictions (*i.e.*, **UC** samples), with prediction going from right to wrong (*i.e.*, **R→W** samples), with predictions going from wrong to right (*i.e.*, **W→R** samples). We analyze the specifics of these three types of samples in detail.

Intuitively, some calibrated prototypes of new classes are aligned to base prototypes and reduce the discriminability of base prototypes. Oppositely, aligning the biased prototype to most similar base prototypes can calibrate the prediction of new classes. As shown in Table 3b, we observe the **W→R** samples are mainly from new classes and the **R→W** samples are mainly from base classes. Besides, extensive comparison results on the benchmark datasets (*i.e.*, Table 3 and Figure 4) show that the negative effect of TEEN is negligible due to the significance of TEEN.

Table 3: Detailed average accuracy of each incremental session on *mini*ImageNet dataset. Please refer to the supplementary for results on CUB200 and CIFAR100. The results of compared methods are cited from [42, 57, 58]. ↑ means higher accuracy is better. ↓ means lower PD is better.

| Method | Accuracy in each session (%) ↑ | | | | | | | | | PD ↓ | Δ PD |
|---|---|---|---|---|---|---|---|---|---|---|---|
| | 0 | 1 | 2 | 3 | 4 | 5 | 6 | 7 | 8 | | |
| iCaRL [35] | 61.31 | 46.32 | 42.94 | 37.63 | 30.49 | 24.00 | 20.89 | 18.80 | 17.21 | 44.10 | **+22.65** |
| EEIL [4] | 61.31 | 46.58 | 44.00 | 37.29 | 33.14 | 27.12 | 24.10 | 21.57 | 19.58 | 41.73 | **+20.28** |
| Rebalancing [18] | 61.31 | 47.80 | 39.31 | 31.91 | 25.68 | 21.35 | 18.67 | 17.24 | 14.17 | 47.14 | **+25.69** |
| TOPIC [42] | 61.31 | 50.09 | 45.17 | 41.16 | 37.48 | 35.52 | 32.19 | 29.46 | 24.42 | 36.89 | **+15.44** |
| Decoupled-NegCosine [26] | 71.68 | 66.64 | 62.57 | 58.82 | 55.91 | 52.88 | 49.41 | 47.50 | 45.81 | 25.87 | **+4.42** |
| Decoupled-Cosine [43] | 70.37 | 65.45 | 61.41 | 58.00 | 54.81 | 51.89 | 49.10 | 47.27 | 45.63 | 24.74 | **+3.29** |
| Decoupled-DeepEMD [56] | 69.77 | 64.59 | 60.21 | 56.63 | 53.16 | 50.13 | 47.79 | 45.42 | 43.41 | 26.36 | **+4.91** |
| CEC [57] | 72.00 | 66.83 | 62.97 | 59.43 | 56.70 | 53.73 | 51.19 | 49.24 | 47.63 | 24.37 | **+2.92** |
| FACT [58] | 72.56 | 69.63 | 66.38 | 62.77 | 60.6 | 57.33 | 54.34 | 52.16 | 50.49 | 22.07 | **+0.62** |
| TEEN | **73.53** | **70.55** | **66.37** | **63.23** | **60.53** | **57.95** | **55.24** | **53.44** | **52.08** | **21.45** | |

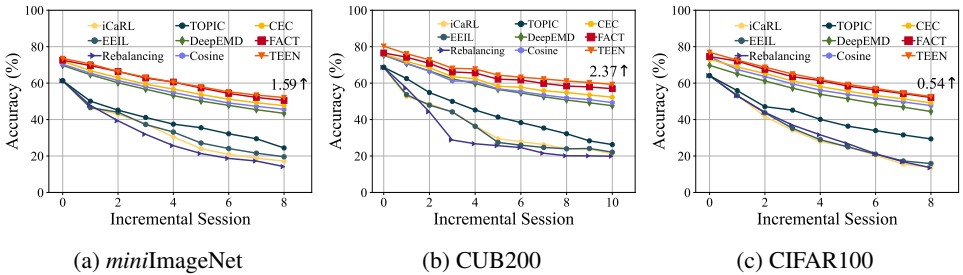

| (a) *mini*ImageNet | (b) CUB200 | (c) CIFAR100 |
|---|---|---|

Figure 4: Top-1 average accuracy on all seen classes in each incremental session. We annotate the performance gap between TEEN and the runner-up method by ↑.

# 6 Experiments

In this section, we first introduce the main experiment details of FSCIL in section 6.1, which include the implementation and performance detail. Subsequently, we introduce the FSL performance of TEEN in section 6.2, which consistently shows the effectiveness of TEEN. Finally, we evaluate TEEN by a comprehensive ablation study in section 6.3.

## 6.1 Main experimental details of FSCIL

### 6.1.1 Implementation Details

**Datasets and baseline details:** Following previous methods [42, 57, 58], we evaluate TEEN on CIFAR100 [23], CUB200-2011 [44], *mini*ImageNet [38]. We keep the dataset split consistent with existing methods [57, 58]. Notably, each benchmark dataset is divided into subsets containing nonoverlapping label space. For example, CIFAR100 is divided into 60 classes for the base session and the left 40 classes are divided into eight 5-way 5-shot few-shot classification tasks. To validate the performance of TEEN , we compare TEEN with popular CIL methods [35, 4, 18], FSL methods [26, 56, 43], and FSCIL methods [42, 57, 58]. Please refer to the supplementary material for more datasets and baseline details.

**Training details:** All experiments are conducted with PyTorch [32] on a single NVIDIA 3090. The training of the feature extractor uses vanilla cross-entropy loss as the objective function. It does not evolve any extra complex pretraining module [58, 59, 10, 57], thus making TEEN more efficient and elegant. In addition, we adopt the cosine similarity to measure the similarity between the instances and class prototypes. Following [42, 57, 58], we use ResNet20 [16] for CIFAR100, pre-trained ResNet18 [16] for CUB200 and randomly initialized ResNet18 [16] for *mini*ImageNet. All compared methods use *the same backbone network and initialization* for a fair comparison. We set $\alpha = 0.5, \tau = 16$ for *mini*ImageNet and CUB200, $\alpha = 0.1, \tau = 16$ for CIFAR100. We train the feature extractor on CUB200 with a learning rate of 0.004, batch size of 128, and epochs of 400. Please refer to the supplementary for more training details on CIFAR100 and *mini*ImageNet.

Table 4: Detailed results of **HMean** and **NAcc** on *mini*ImageNet. The best results are in bold and the runner-up results are in underlines. The $\Delta$ measures the performance gap between the best and second-best results on the corresponding session. Due to space limitations, the performance on only six incremental sessions is presented. Please refer to the supplementary for more detailed results on CUB200 and CIFAR100.

| Session | 1 | | 2 | | 3 | | 4 | | 5 | | 6 | |
|---|---|---|---|---|---|---|---|---|---|---|---|---|
| **HMean/NAcc** | HMean | NAcc | HMean | NAcc | HMean | NAcc | HMean | NAcc | HMean | NAcc | HMean | NAcc |
| CEC [57] | 30.72 | 19.60 | 30.05 | 19.10 | 29.86 | 19.00 | 29.41 | 18.65 | 27.15 | 16.88 | 27.36 | 17.07 |
| FACT [58] | 30.60 | 19.20 | 27.84 | 17.10 | 25.89 | 15.67 | 23.85 | 14.20 | 22.01 | 12.92 | 20.65 | 12.00 |
| TEEN | **50.04** | **38.00** | **46.67** | **34.60** | **44.72** | **32.67** | **43.53** | **31.55** | **41.75** | **29.80** | **39.22** | **27.37** |
| $\Delta$ | +19.32 | +18.4 | +16.62 | +15.5 | +14.86 | +13.67 | +14.12 | +12.9 | +14.6 | +12.92 | +11.86 | +10.3 |

### 6.1.2 Comparison results

In this section, we conduct overall comparison experiments on different performance measures. These different performance measures focus on different aspects of the methods. For example, the widely-used average accuracy across all classes (*i.e.*, **AvgAcc**) is difficult to reflect the performance of new classes because the base classes take a large percentage of all classes (*e.g.*, 60 base classes of 100 classes in CIFAR100). Following [58], the Harmonic mean (*i.e.*, **HMean**) is used to evaluate the balanced performance between the base and new classes. Besides the above performance measures, we also additionally evaluate the different methods of their performance on new classes (*i.e.*, **NAcc**). Following [42, 57, 58], we also measure the degree of forgetting by the **P**erformance **D**ropping Rate (*i.e.*, **PD**). The PD is defined as $PD = Acc_0 - Acc_{-1}$, *i.e.*, the average accuracy dropping between the first session (*i.e.*, $Acc_0$) and the last session (*i.e.*, $Acc_{-1}$). The detailed comparison results of **PD** and **AvgAcc** are reported in Table 3, and the detailed comparison results of **NAcc** and **HMean** are reported in Table 4. These experimental results from different performance measures all demonstrate the effectiveness of TEEN. Besides, significantly lower FPR and TBR in Table 1 and Table 2 also verify the effective calibration of TEEN.

Notably, many previous state-of-the-art FSCIL methods (*e.g.*, [57, 58]) usually design complex pretraining algorithms to enhance the extendibility of feature space, which may harm the discriminability of base classes. This elaborate pretraining stage may lead to an inconsistent but *negligible* performance in the base session. Besides, PD↓ and $\Delta$PD in Table 3 are not affected by the pretrained results and also show TEEN outperforms previous state-of-the-art methods.

## 6.2 Comparison results of FSL

FSL can be approximated as measuring only the performance of the new classes in the first incremental session of FSCIL. Besides, FSL itself also faces the challenge of biased class prototypes due to the few-shot data. Therefore, we validate TEEN in the setting of FSL and report the compared results in Table 5. To ensure a fair comparison, we strictly followed the experimental setup of [51]. The comparison results in Table 5 show

Table 5: Few-Shot Leaning performance of classification accuracy (%) on *mini*ImageNet and CUB. The results of compared methods are cited from [51]. 5w1s and 5w5s mean 5way-1shot and 5way-5shot, respectively. The best results are in bold and the runner-up results are in underlines.

| Methods | *mini*ImageNet | | CUB | |
|---|---|---|---|---|
| | 5w1s | 5w5s | 5w1s | 5w5s |
| ProtoNet [40] | 54.16 | 73.68 | 72.99 | 86.64 |
| NegCosine [26] | 62.33 | 80.94 | 72.66 | 89.40 |
| LR with DC [51] | **68.57** | 82.88 | 79.56 | 90.67 |
| TEEN | 65.70 | **83.11** | **81.44** | **91.04** |

that TEEN can easily *outperform the previous state-of-the-art method* [51] in several experimental settings. Notably, TEEN *does not involve any additional training cost when recognizing new classes*, and the only time cost lies in feature extraction. Therefore, once the sample features are extracted, **the inference cost of** TEEN **can be considered negligible** compared to previous methods that require heavy training.

## 6.3 Ablation Study

**The influence of $\alpha$ and $\tau$:** Notably, the proposed TEEN does not involve additional training-based modules or procedures after pretraining on base classes. When TEEN incrementally learns new classes, only the scaling temperature $\tau$ and the coefficient of calibration item $\alpha$ need to be determined.

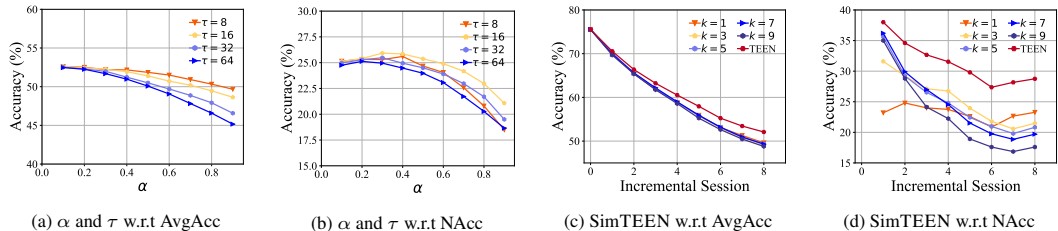

| (a) $\alpha$ and $\tau$ w.r.t AvgAcc | (b) $\alpha$ and $\tau$ w.r.t NAcc | (c) SimTEEN w.r.t AvgAcc | (d) SimTEEN w.r.t NAcc |

Figure 5: (a) and (b) show the influence of $\alpha$ and $\tau$ on the last session's average accuracy in *all classes* (*i.e.*, AvgAcc) and *new classes* (*i.e.*, NAcc), respectively. (c) and (d) show the influence of similarity-based weight in TEEN on the last session's average accuracy in *all classes* (*i.e.*, AvgAcc) and *new classes* (*i.e.*, NAcc), respectively. It demonstrates that replacing the softmax-based weight with a hard one-shot weight will drastically reduce the performance of TEEN.

Specifically, we select $\tau$ from $\{8, 16, 32, 64\}$ and $\alpha$ from $\{0.1, 0.2, 0.3, 0.4, 0.5, 0.6, 0.7, 0.8, 0.9\}$. Figure 5a and Figure 5b show how the hyper-parameters $\alpha$ and $\tau$ influence the average accuracy on all classes and new classes, respectively. It also demonstrates that a proper $\alpha$ can improve the performance of new classes, which confirms the base prototypes can benefit the performance of new classes. Compared with larger $\tau$, relatively smaller $\tau$ can smooth the weight distribution. The smaller $\tau$ with stronger performance further confirm the effectiveness of utilizing the abundant semantic information from base classes.

**The influence of similarity-based weight:** To further validate the effectiveness of the semantic similarity between the base and new classes, we remove the Equation 6 and directly align the new prototypes to their corresponding $K$ most similar base class prototypes, *i.e.*, $\bar{\mathbf{c}}_n = \alpha\,\mathbf{c}_n + (1 - \alpha)\sum_{k=1}^{K}\mathbf{c}_k$. We denote this **Sim**ple version of TEEN *without similarity-based weight* as SimTEEN. We select $K$ from $\{1, 3, 5, 7, 9\}$. Figure 5c and Figure 5d show how the similarity-based weight (*i.e.*, Equation 6) influences the average accuracy on all classes and new classes, respectively. Notably, as shown in Figure 5d, there is a significant drop, particularly on new classes, after removing the similarity-based weight. This phenomenon confirms that TEEN can achieve calibration of new prototypes with the help of well-represented semantic similarity between the base and new classes.

## 7  Conclusion

Few-shot class-incremental learning is of great importance to real-world learning scenarios. In this study, we first observe existing methods usually exhibit poor performance in new classes and the samples of new classes are easily misclassified into most similar base classes. We further find that although the feature extractor trained on base classes can not well represent new classes directly, it can properly represent the semantic similarity between the base and new classes. Based on these analyses, we propose a simple yet effective training-free prototype calibration strategy (*i.e.*, TEEN) for biased prototypes of new classes. TEEN obtains competitive results in FSCIL and FSL scenarios.

**Limitations:** The FSCIL methods mentioned in this paper all select the base and new classes from the *same* dataset. In other words, current FSCIL methods assume the model pre-trains in the same domain, increasing the restrictions on pre-training data collection. Therefore, a more realistic scenario is to pre-train on a dataset that is independent of the subsequent data distribution and then perform few-shot class-incremental learning on a target dataset that we expect. The intricate problem of cross-domain few-shot class-incremental learning will be thoroughly investigated in our future works.

## Acknowledgements

This work is partially supported by National Key R&D Program of China (2022ZD0114805), NSFC (62376118, 62006112, 62250069), Young Elite Scientists Sponsorship Program of Jiangsu Association for Science and Technology 2021-020, Collaborative Innovation Center of Novel Software Technology and Industrialization.

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

# Supplementary materials

## A  Discussion

### A.1  Implicit Semantic Similarity

Notably, TEEN demonstrates that *the off-the-shelf feature extractor* is already capable of representing semantic similarity without the need for any additional training costs or auxiliary side information (*e.g.*, the textual information of the class name [8]).

### A.2  Efficiency

The only time cost in TEEN is the pre-training cost on the base classes. Furthermore, the training cost of the vanilla cross-entropy loss function is relatively small compared to other more complex training paradigms. Additionally, the calibration in TEEN does not evolve any training cost and update procedure. Therefore, the cost of incrementally learning new classes of TEEN can be considered negligible compared with existing methods.

### A.3  Combination with better representation

In addition to combining with existing prototype-based methods, TEEN can also be combined with more powerful pre-trained models (*e.g.*, self-supervised pre-trained model [7]). Notably, the recent popular vision-language models (*e.g.*, CLIP [34]) can also be seen as a powerful pre-trained model and provide better representation.

## B  A Closer Look at FSCIL

In this section, we show the observations (*i.e.*, **Section 3** of the main paper) on more datasets. As shown in Table 10, the existing prototype-based FSCIL methods are extremely prone to misclassify new classes into base classes. Considering the misclassified instances of base classes, we quantitatively show that *the samples of new classes are usually misclassified into base classes*. These observations in the additional benchmark dataset are consistent with observations in the main paper, which further confirms the universality of our observation.

To further understand the aforementioned observation, we further analyze the question *What base classes are the new classes incorrectly predicted into?* and come to our second observation: **The prototype-based classifier misclassifies the new classes to their corresponding most similar base classes with a high probability.** The detailed analysis results in Table 11 further verify the correctness of our observation.

Table 6: Detailed average accuracy of each incremental session on CIFAR100 dataset.The results of compared methods are cited from [42, 57, 58]. ↑ means higher accuracy is better. ↓ means lower PD is better.

| Method | Accuracy in each session (%) ↑ | | | | | | | | | PD ↓ | Δ PD |
|---|---|---|---|---|---|---|---|---|---|---|---|
| | 0 | 1 | 2 | 3 | 4 | 5 | 6 | 7 | 8 | | |
| iCaRL [35] | 64.10 | 53.28 | 41.69 | 34.13 | 27.93 | 25.06 | 20.41 | 15.48 | 13.73 | 50.37 | **+28.09** |
| EEIL [4] | 64.10 | 53.11 | 43.71 | 35.15 | 28.96 | 24.98 | 21.01 | 17.26 | 15.85 | 48.25 | **+25.97** |
| Rebalancing [18] | 64.10 | 53.05 | 43.96 | 36.97 | 31.61 | 26.73 | 21.23 | 16.78 | 13.54 | 50.56 | **+28.28** |
| TOPIC [42] | 64.10 | 55.88 | 47.07 | 45.16 | 40.11 | 36.38 | 33.96 | 31.55 | 29.37 | 34.73 | **+12.45** |
| Decoupled-NegCosine [26] | 74.36 | 68.23 | 62.84 | 59.24 | 55.32 | 52.88 | 50.86 | 48.98 | 46.66 | 27.70 | **+5.42** |
| Decoupled-Cosine [43] | 74.55 | 67.43 | 63.63 | 59.55 | 56.11 | 53.80 | 51.68 | 49.67 | 47.68 | 26.87 | **+4.59** |
| Decoupled-DeepEMD [56] | 69.75 | 65.06 | 61.20 | 57.21 | 53.88 | 51.40 | 48.80 | 46.84 | 44.41 | 25.34 | **+3.06** |
| CEC [57] | 73.07 | 68.88 | 65.26 | 61.19 | 58.09 | 55.57 | 53.22 | 51.34 | 49.14 | 23.93 | **+1.65** |
| FACT [58] | 74.60 | 72.09 | 67.56 | 63.52 | 61.38 | 58.36 | 56.28 | 54.24 | 52.10 | 22.50 | **+0.22** |
| TEEN | **74.92** | **72.65** | **68.74** | **65.01** | **62.01** | **59.29** | **57.90** | **54.76** | **52.64** | **22.28** | |

Table 7: Detailed average accuracy of each incremental session on CUB200 dataset. The results of compared methods are cited from [42, 57, 58]. ↑ means higher accuracy is better. ↓ means lower PD is better.

| Method | Accuracy in each session (%) ↑ | | | | | | | | | | | PD ↓ | Δ PD |
|---|---|---|---|---|---|---|---|---|---|---|---|---|---|
| | 0 | 1 | 2 | 3 | 4 | 5 | 6 | 7 | 8 | 9 | 10 | | |
| iCaRL [35] | 68.68 | 52.65 | 48.61 | 44.16 | 36.62 | 29.52 | 27.83 | 26.26 | 24.01 | 23.89 | 21.16 | 47.52 | +29.39 |
| EEIL [4] | 68.68 | 53.63 | 47.91 | 44.20 | 36.30 | 27.46 | 25.93 | 24.70 | 23.95 | 24.13 | 22.11 | 46.57 | +28.44 |
| Rebalancing [18] | 68.68 | 57.12 | 44.21 | 28.78 | 26.71 | 25.66 | 24.62 | 21.52 | 20.12 | 20.06 | 19.87 | 48.81 | +30.68 |
| TOPIC [42] | 68.68 | 62.49 | 54.81 | 49.99 | 45.25 | 41.40 | 38.35 | 35.36 | 32.22 | 28.31 | 26.26 | 42.40 | +24.27 |
| Decoupled-NegCosine† [26] | 74.96 | 70.57 | 66.62 | 61.32 | 60.09 | 56.06 | 55.03 | 52.78 | 51.50 | 50.08 | 48.47 | 26.49 | +7.36 |
| Decoupled-Cosine [43] | 75.52 | 70.95 | 66.46 | 61.20 | 60.86 | 56.88 | 55.40 | 53.49 | 51.94 | 50.93 | 49.31 | 26.21 | +8.08 |
| Decoupled-DeepEMD [56] | 75.35 | 70.69 | 66.68 | 62.34 | 59.76 | 56.54 | 54.61 | 52.52 | 50.73 | 49.20 | 47.60 | 27.75 | +9.62 |
| CEC [57] | 75.85 | 71.94 | 68.50 | 63.50 | 62.43 | 58.27 | 57.73 | 55.81 | 54.83 | 53.52 | 52.28 | 23.57 | +5.44 |
| FACT [58] | 75.90 | 73.23 | 70.84 | 66.13 | 65.56 | 62.15 | 61.74 | 59.83 | 58.41 | 57.89 | 56.94 | 18.96 | +0.83 |
| TEEN | **77.26** | **76.13** | **72.81** | **68.16** | **67.77** | **64.40** | **63.25** | **62.29** | **61.19** | **60.32** | **59.31** | **18.13** | |

Table 8: Detailed results of **HMean** and **NAcc** on *mini*ImageNet. The best results are in bold and the runner-up results are in underlines. The Δ measures the performance gap between the best and second-best results on the corresponding session. Due to space limitations, the performance on only six incremental sessions is presented. Please refer to the supplementary for more detailed results on CUB200 and CIFAR100.

| Session | 1 | | 2 | | 3 | | 4 | | 5 | | 6 | | 7 | | 8 | |
|---|---|---|---|---|---|---|---|---|---|---|---|---|---|---|---|---|
| **HMean/NAcc** | HMean | NAcc | HMean | NAcc | HMean | NAcc | HMean | NAcc | HMean | NAcc | HMean | NAcc | HMean | NAcc | HMean | NAcc |
| CEC [57] | 47.04 | 34.80 | 40.35 | 28.10 | 36.01 | 24.13 | 36.49 | 24.65 | 36.56 | 24.84 | 37.07 | 25.40 | 36.41 | 24.83 | 35.49 | 24.08 |
| FACT [58] | 43.04 | 30.00 | 38.77 | 26.10 | 34.70 | 22.60 | 34.44 | 22.45 | 34.55 | 22.64 | 35.44 | 23.47 | 33.79 | 22.06 | 33.05 | 21.48 |
| TEEN | 47.77 | 34.82 | 42.99 | 30.10 | 40.19 | 27.53 | 39.52 | 27.00 | 39.66 | 27.28 | 39.65 | 27.37 | 37.71 | 25.57 | 37.11 | 25.13 |
| Δ | +0.73 | +0.02 | +2.64 | +2.00 | +4.18 | +3.40 | +3.03 | +2.35 | +3.10 | +2.44 | +2.58 | +1.97 | +1.30 | +0.74 | +1.62 | +1.05 |

# C Experiments

## C.1 Details of experiments

**Dataset details:** Following previous methods [42, 57, 58], we evaluate TEEN on CIFAR100 [23], CUB200-2011 [44], *mini*ImageNet [38]. CIFAR100 [23] contains 100 classes and each class contains 500 images for training and 100 images for testing. The image size of CIFAR100 is $3 \times 32 \times 32$. The CUB200-2011 [44] is a widely-used fine-grained dataset and is also a benchmark dataset of few-shot image classification. It contains 200 classes and all images of these classes belong to birds. The *mini*ImageNet [38] are sampled from the raw ImageNet [11] and contains 100 classes.

In FSCIL, each benchmark dataset is divided into different subsets. Each subset contains specific classes and the label space of different subsets is nonoverlapping. Specifically, CIFAR100 is divided into 60 classes for the base session and the remaining 40 classes are divided into eight 5-way 5-shot few-shot classification tasks. CUB200 is divided into 100 base classes for the base session and the remaining 100 classes are divided into ten 10-way 5-shot few-shot classification tasks. *mini*ImageNet is divided into 60 base classes for the base session, and the remaining 40 classes are divided into eight 5-way 5-shot few-shot classification tasks. The splitting details (*i.e.*, the class order and the selection of support data in incremental sessions) follow the previous methods [42, 57, 58]. Notably, **no old samples are saved** to assist in maintaining the discriminability of previous classes.

**Baseline details:** Following previous methods [42, 57, 58, 61], we compare TEEN with popular CIL methods, FSL methods, and FSCIL methods. For CIL methods, we select iCaRL [35], EEIL [4] and Rebalancing [18] as our baseline methods. For methods based on few-shot, we adopt Decoupled-NegativeCosine [26], Decoupled-Cosine [43] and Decoupled-DeepEMD [56] as our baseline methods. For FSCIL methods, we adopt TOPIC [42], CEC [57] and FACT [58] methods as our baseline methods.

**Training details:** The training of the feature extractor uses vanilla cross-entropy loss as the objective function. In addition, we adopt the cosine similarity to measure the feature of instances to class prototypes. Following [42, 57, 58], we use ResNet20 [16] for CIFAR100, pre-trained ResNet18 [16] for CUB200 and randomly initialized ResNet18 [16] for *mini*ImageNet. All compared methods use **the same backbone network and same initialization** for a fair comparison.

For the hyperparameters setting, we set $\alpha = 0.5, \tau = 16$ for *mini*ImageNet and CUB200, $\alpha = 0.1, \tau = 16$ for CIFAR100. We train the feature extractor on CUB200 with a learning rate of 0.004,

batch size of 128, and epochs of 400. We train the feature extractor on CIFAR100 and *mini*ImageNet with a learning rate of 0.1 and batch size of 256. The cosine scheduler is used to adjust the learning rate. Before computing the cross-entropy loss, a widely-used temperature scalar is used to adjust the distribution of output logits. For example, the original output logits of instance $x_j$ are denoted as $\mathcal{O}_j \in \mathbb{R}^C$. The logits used to compute cross-entropy loss are denoted as $\mathcal{O}_j \tau_o$. The $\tau_o$ is set to 16 for the CIFAR100 dataset and 32 for *mini*ImageNet and CUB200 datasets.

## C.2   More comparison results

In the main paper, we compare different methods in overall performance measures: performance drop rate, average accuracy across all classes, harmonic mean accuracy and average accuracy on new classes. In this session, we report the detailed comparison results of the aforementioned performance measures on CIFAR100 [23] and CUB200 [44]. Besides the main results in Table 6 and Table 7, we also report the accuracy across new classes and harmonic mean accuracy in Table 8. The consistent performance improvement demonstrates the effectiveness of TEEN's calibration ability.

# D   Empirical Analysis

## D.1   Effects of TEEN

In the main paper, we only show the effect of calibration prototypes on the CIFAR100 dataset. In this supplementary material, we report more detailed results in Table 9. The results further confirm our conclusion in the main paper: the proposed TEEN can well calibrate the biased prototype of new classes. Although TEEN leads to misclassifying some instances of base classes, the average accuracy across all classes is always very competitive.

Table 9: The detailed ratio (%) of base and new classes about three types of samples(*i.e.*, **UC** samples, **W→R** samples, **R→W** samples) of *mini*ImageNet

| Incremental Session | 1 | | | 2 | | | 3 | | | 4 | | |
|---|---|---|---|---|---|---|---|---|---|---|---|---|
| Sample type | UC | W→R | R→W | UC | W→R | R→W | UC | W→R | R→W | UC | W→R | R→W |
| Base Class | 93.84 | 0.00 | **99.99** | 88.87 | 0.00 | **95.78** | 84.15 | 0.00 | **94.20** | 79.65 | 0.00 | **91.03** |
| New Class | 6.16 | **100.00** | 0.01 | 11.13 | **100.00** | 4.22 | 15.85 | **100.00** | 5.80 | 20.35 | **100.00** | 8.97 |

## D.2   Combination of TEEN with existing methods

Due to the flexibility of proposed TEEN , it can be seen as a *plug-and-play* module to calibrate the biased prototypes of new classes. For a more in-depth understanding of the TEEN , we combine TEEN with FACT [58] and show the performance on different performance measures in Figure 6, Figure 7 and Figure 8. We empirically find that combining the TEEN with FACT has little effect on the average accuracy across all classes (*i.e.*, the left figures in Figure 6, Figure 7 and Figure 8). However, the TEEN can improve the average accuracy of new classes and the harmonic mean accuracy. In other words, the existing prototype-based methods (*i.e.*, FACT) benefit the well-calibration effect of TEEN and achieve better-balanced performance between the base and new classes. We believe more methods (*e.g.*, [52]) can benefit from the flexibility of TEEN.

# E   Broader impact

Our proposed method can significantly improve the accuracy of new classes in FSCIL and FSL scenarios, leading to a better understanding of the low performance of new classes. Furthermore, *the analyses in this study are first proposed in the FSCIL and FSL fields, which point out that we should pay more attention to the performance of new classes*. In addition to this, our method may also have potential heuristic effects on the expectation estimation of the distribution.

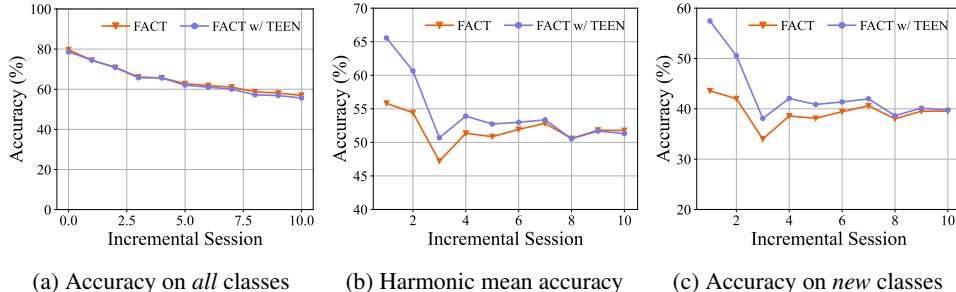

(a) Accuracy on *all* classes     (b) Harmonic mean accuracy     (c) Accuracy on *new* classes

Figure 6: We compare FACT without (*i.e.*, **FACT** in figures) and with TEEN (*i.e.*, **FACT w/** TEEN on CUB200 dataset. The FACT benefits from the well-calibration effect of TEEN and achieves better-balanced performance between the base and new classes

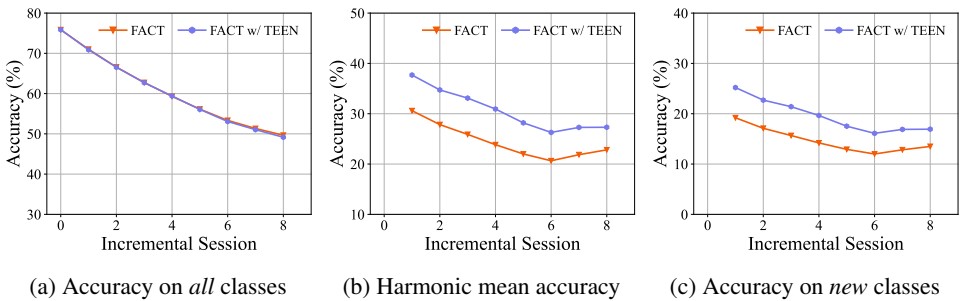

(a) Accuracy on *all* classes     (b) Harmonic mean accuracy     (c) Accuracy on *new* classes

Figure 7: We compare FACT without (*i.e.*, **FACT** in figures) and with TEEN (*i.e.*, **FACT w/** TEEN on *mini*ImageNet dataset. The FACT benefits from the well-calibration effect of TEEN and achieves better-balanced performance between the base and new classes

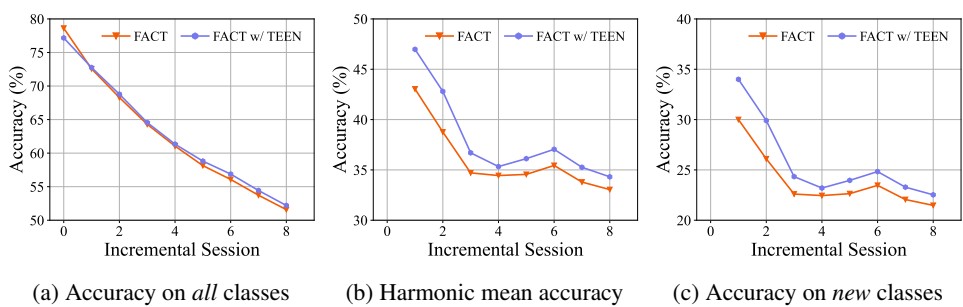

(a) Accuracy on *all* classes     (b) Harmonic mean accuracy     (c) Accuracy on *new* classes

Figure 8: We compare FACT without (*i.e.*, **FACT** in figures) and with TEEN (*i.e.*, **FACT w/** TEEN on CIFAR100 dataset. The FACT benefits from the well-calibration effect of TEEN and achieves better-balanced performance between the base and new classes

Table 10: Detailed prediction results of **False Negative Rate/False Positive Rate** (%) on *mini*ImageNet dataset. The analysis results are from session 1 because new classes do not exist in session 0. Exceedingly high **FPR** and relatively low **FNR** show the instances of new classes are easily misclassified into base classes and the instances of base classes are also easily misclassified into base classes. TEEN can achieve relatively lower **FPR** than baseline methods, which demonstrates the validity of the proposed calibration strategy.

| Session | | 1 | | 2 | | 3 | | 4 | | 5 | | 6 | | 7 | | 8 |
|---|---|---|---|---|---|---|---|---|---|---|---|---|---|---|---|---|
| **FNR/FPR** | FNR | FPR | FNR | FPR | FNR | FPR | FNR | FPR | FNR | FPR | FNR | FPR | FNR | FPR | FNR | FPR |
| ProtoNet [40] | 2.55 | 71.60 | 3.98 | 67.30 | 4.83 | 66.07 | 5.83 | 62.05 | 6.48 | 59.16 | 7.18 | 58.17 | 7.93 | 54.83 | 8.85 | 52.23 |
| CEC [57] | 3.45 | 68.40 | 5.60 | 65.50 | 7.03 | 61.93 | 7.73 | 58.30 | 8.47 | 56.68 | 9.58 | 54.50 | 10.22 | 52.54 | 10.93 | 50.05 |
| FACT [58] | 2.07 | 72.40 | 3.42 | 70.60 | 4.22 | 70.13 | 4.55 | 69.40 | 4.97 | 68.28 | 5.12 | 68.17 | 5.33 | 66.51 | 5.58 | 66.55 |
| TEEN | 8.02 | **46.40** | 11.35 | **38.60** | 13.12 | **37.53** | 15.32 | **35.20** | 16.47 | **32.48** | 17.38 | **31.57** | 18.63 | **28.03** | 19.97 | **26.35** |

Table 11: Detailed prediction results of **TNR/TBR** (%) on *mini*ImageNet dataset. The analysis results are from session 1 because new classes do not exist in session 0. For new classes, we only consider the 10 most similar base classes out of 60 base classes. For base classes, we suppose $C_i$ new classes exist in the current incremental session $i$. We only consider the most similar $\lfloor 20\% \times C_i \rfloor$ new classes. Class similarity adopts cosine similarity between different class prototypes. TEEN can achieve relatively lower **TBR** than baseline methods, which demonstrates the validity of the proposed calibration strategy.

| Session | 1 | | 2 | | 3 | | 4 | | 5 | | 6 | | 7 | | 8 | |
|---|---|---|---|---|---|---|---|---|---|---|---|---|---|---|---|---|
| **TNR/TBR** | TNR | TBR | TNR | TBR | TNR | TBR | TNR | TBR | TNR | TBR | TNR | TBR | TNR | TBR | TNR | TBR |
| ProtoNet [40] | 3.47 | 73.01 | 5.02 | 61.79 | 7.06 | 55.05 | 8.08 | 52.95 | 7.14 | 53.27 | 8.56 | 51.96 | 8.53 | 49.18 | 9.21 | 50.91 |
| CEC [57] | 2.87 | 71.70 | 4.53 | 61.08 | 5.66 | 58.26 | 6.35 | 55.16 | 6.07 | 54.12 | 6.92 | 52.53 | 8.24 | 49.96 | 8.20 | 51.64 |
| FACT [58] | 2.33 | 70.00 | 3.32 | 61.01 | 5.50 | 55.17 | 6.88 | 50.55 | 6.57 | 50.52 | 7.72 | 49.30 | 8.88 | 46.99 | 9.31 | 48.48 |
| TEEN | 2.16 | **66.77** | 2.53 | **55.14** | 3.55 | **44.87** | 3.85 | **37.68** | 3.57 | **39.47** | 4.02 | **37.07** | 4.76 | **33.83** | 4.85 | **35.04** |

