# OpenReview forum: "Few-Shot Class-Incremental Learning via Training-Free Prototype Calibration"
_NeurIPS.cc/2023/Conference — NeurIPS 2023 poster_

### Official Review · Reviewer_TNew · 2023-06-19

**Soundness:** 3 good
**Presentation:** 3 good
**Contribution:** 3 good
**Rating:** 5
**Confidence:** 4

**Summary:**

The paper proposes a strategy called Training-frEE calibratioN (TEEN) for Few-Shot Class-Incremental Learning (FSCIL) scenario to enhance the discriminability of new classes by fusing the new prototypes with weighted base prototypes. TEEN demonstrates remarkable performance and consistent improvements over baseline methods in the few-shot learning scenario.

**Strengths:**

The paper addresses the Few-Shot Class-Incremental Learning (FSCIL) scenario, which is a challenging and important problem in real-world scenarios.

The proposed strategy, TEEN, is simple yet effective and demonstrates remarkable performance and consistent improvements over baseline methods in the few-shot learning scenarios.


**Weaknesses:**

The paper demonstrates significant improvement on new classes. However, based on the experimental results, it appears that the False Negative ratio, which involves classifying base instances into incorrect classes, may increase. It would be beneficial to address how this problem is handled and how the performance can be balanced between base classes and new classes. Additional analysis on this topic would be appreciated.

As stated by the authors, pre-training on a dataset that is independent of the subsequent data distribution would be advantageous.

The paper lacks a comparison of the proposed method with state-of-the-art methods in other few-shot learning scenarios, such as few-shot domain adaptation or few-shot semi-supervised learning. It would be beneficial to include more analysis and comparisons with these methods.

**Questions:**

See the weakness section for more details

**Limitations:**


See the weakness section for more details

---

> ### Author Rebuttal · Authors · 2023-08-10
>
> **Q1**  The paper demonstrates significant improvement on new classes. However, based on the experimental results, it appears that the False Negative ratio, which involves classifying base instances into incorrect classes, may increase. It would be beneficial to address how this problem is handled and how the performance can be balanced between base classes and new classes. Additional analysis on this topic would be appreciated
>
> **A1** We thank the reviewer for the constructive question. Firstly, we comprehensively analyze the potential negative effect of TEEN in section 4.2. From the final benchmark results presented in Section 5, it can be observed that **the potential negative effect is overshadowed by the more substantial positive effect of the improvement in the new prototypes**. Furthermore, we include a hyper-parameter $\alpha$ to adjust the strength of the calibration. We give a detailed ablation study of $\alpha$ in Figure 5. The hyper-parameter $\alpha$ temporarily provides a simple solution to ensure performance and more complicated methods to mitigate the increasing False Negative ratio may be designed in the future.
>
> ---
>
> **Q2** As stated by the authors, pre-training on a dataset that is independent of the subsequent data distribution would be advantageous. The paper lacks a comparison of the proposed method with state-of-the-art methods in other few-shot learning scenarios, such as few-shot domain adaptation or few-shot semi-supervised learning. It would be beneficial to include more analysis and comparisons with these methods.
>
> **A2** We thank the reviewer for the suggestion and the potentially related field. To the best of our knowledge, conventional methods in the few-shot scenarios do not consider the process of *incremental learning.* These methods were not designed with the continuous arrival of new data in mind, **making them unsuitable for FSCIL tasks**. If there are any papers closely related to the cross-dataset few-shot class-incremental learning scenario, we would appreciate it if you could directly point out those relevant papers and we will discuss these papers in the final version. Additionally, there is currently no strict definition for cross-dataset few-shot class-incremental learning (e.g., the selection of datasets or choice of evaluation metrics). In fact, we consider the cross-dataset setup as a general assumption and limitation in all FSCIL problems. Hence, in the **limitations** section, we mention this as a potential future research direction. However, to better demonstrate the performance of TEEN, we attempted a simple cross-dataset experiment by constructing a **CUB200->miniImageNet** dataset comparison to address your potential concerns. The results are shown in the global response (**Cross-dataset  few-shot class-incremental learning scenario**). Notably, TEEN still shows competitive performance. We speculate that existing methods (e.g., CEC and FACT) often employ complex learning algorithms that lead to excessive adaptation to the pre-training dataset. In contrast, TEEN relies on the calibration of semantic similarity, which may alleviate the potential negative impact of dataset shift.

---

### Official Review · Reviewer_3Zyg · 2023-07-01

**Soundness:** 3 good
**Presentation:** 3 good
**Contribution:** 2 fair
**Rating:** 5
**Confidence:** 5

**Summary:**

This paper tackles the problem of few-shot class incremental learning based on prototypical network. Motivated by the observation that novel classes are easily misclassified as base classes, the authors propose a prototype calibration strategy. The calibrated prototype is a weighted sum of prototype computed from novel class support set and base classes semantically similar with the novel class. Experiments on miniImageNet, CUB and CIFAR demonstrate the superiority of the proposed method over previous ones.

**Strengths:**

1. The paper is well motivated that the inferior performance of prototypical network results from the phenomenon that novel classes are easily misclassified as base classes. The phenomenon is well-studied via experiments.
2. The paper is well written and easy to follow.
3. The authors propose a simple but effective calibration strategy to improve the performance.

**Weaknesses:**

1. The paper introduces two hyper parameters which need to be exhaustively searched for every dataset.
2. There lacks cross-dataset experiments to show the effectiveness of the proposed method when transferring knowledge from one dataset to another.

**Questions:**

The semantic similarity based calibration is a straightforward and commonly used method used in the classification area. There may lack technical contribution to the community.

**Limitations:**

The limitation of cross-dataset evaluation is discussed.

---

> ### Author Rebuttal · Authors · 2023-08-10
>
> **Q1** The paper introduces two hyper parameters which need to be exhaustively searched for every dataset.
>
> **A1** We thank the reviewer for the constructive question. We show the performance trend with respect to $\alpha$ and $\tau$ on each benchmark dataset in   **Figure 2 in the Rebuttal PDF**. To better illustrate the trends, we have included the replicated trend in figure 5 (a)) in this rebuttal pdf.  We observed that the trend of the hyperparameter $\alpha$ remains consistent across all benchmark datasets. Although the trend of the hyperparameter $\tau$ may vary slightly between the CUB200, CIFAR100, and miniImageNet datasets, maintaining uniform settings still allows TEEN to maintain competitive performance. Notably, the trend of $\tau$ in miniImageNet and CIFAR100 datasets is identical. We hypothesize that the slight difference observed in CUB200 may be attributed to the fact that it is a *fine-grained* dataset.
>
> ---
>
> **Q2** There lacks cross-dataset experiments to show the effectiveness of the proposed method when transferring knowledge from one dataset to another.
>
> **A2**  In fact, we introduce this more realistic FSCIL scenario (e.g., pre-training and fine-tuning on *independent* datasets) in our **limitation** part. We believe that this is a limitation currently present in mainstream FSCIL settings and could be a potential research direction. This limitation can be viewed as an assumed constraint in all existing FSCIL methods, which can be further explored in future research. Nevertheless, we construct a more strict FSCIL scenario to evaluate the effectiveness of TEEN. Please refer to the global response (**Cross-dataset  few-shot class-incremental learning scenario**) for detailed results for this part. Specifically, we pre-train the model on the base classes of the CUB200 dataset and incrementally learn the few-shot tasks in the miniImageNet dataset. The comparison results show that TEEN can also perform competitively. We speculate that existing methods often overly rely on specialized learning processes on the pre-training dataset. For example, methods like CEC require learning a classiﬁer adaptation module on the base classes. These modules, which are excessively trained on the pre-training dataset, may result in poor performance of existing methods in incremental learning scenarios with dataset shift. In contrast, *TEEN does not introduce additional modules for adaptation*. It simply leverages the potential semantic similarity between classes to calibrate the prototypes. This training-free characteristics allow TEEN to maintain competitive performance in scenarios with dataset shifts.
>
> ---
>
> **A3** The semantic similarity based calibration is a straightforward and commonly used method used in the classification area. There may lack technical contribution to the community.
>
> **Q3** Different from existing semantic-based calibration methods, we propose a specific *training-free* calibration method (TEEN) to improve the discriminability of novel classes. Notably, the feature extractor used in TEEN is *only* trained on base classes and *does not involve any extra module or data to characterize the semantic similarity*. Besides, the empirical analysis and observations in our paper are also meaningful in the scenario of FSCIL. In our global response, we reiterate the contributions and novelty of our paper. Please refer to the global response for a detailed conclusion.

---

> ### Comment · Reviewer_3Zyg · 2023-08-16
>
> The rebuttal resolves my concerns and I would like to keep my original score (5: Borderline accept).

---

> > ### Author Response · Authors · 2023-08-19
> > **Response to 3Zyg**
> >
> > We greatly appreciate your support and welcome further discussion if you have any additional questions or concerns.

---

### Official Review · Reviewer_aEVg · 2023-07-05

**Soundness:** 2 fair
**Presentation:** 2 fair
**Contribution:** 2 fair
**Rating:** 5
**Confidence:** 4

**Summary:**

The authors work on the Few-Shot Class-Incremental Learning (FSCIL) scenario and propose the Training-frEE calibratioN (TEEN) strategy. This strategy enhances the discriminability of new classes by fusing the new prototypes with base prototypes. This approach is different from previous methods, which either introduce extra learnable components or rely on a frozen feature extractor.

**Strengths:**

1. The authors observed that the feature extractor, although only trained on base classes, can surprisingly represent the semantic similarity between the base and unseen new classes. Accordingly, they proposed a solution.
2. The paper is easy to follow, with clear experimental details that aid reproducibility. The motivations are also presented clearly.

**Weaknesses:**

1. The issue of misclassification has already been observed and studied in previous few-shot learning works. Therefore, the poor performance of new classes is not surprising in the task of Few-Shot Class-Incremental Learning (FSCIL).
2. Forming connections across samples from base and novel classes is not a new concept. For instance, FADI[1] discovered that a novel class may implicitly leverage the knowledge of multiple base classes to construct its feature space. It then builds a discriminative feature space for each novel class via association and discrimination steps.
[1] Few-Shot Object Detection via Association and Discrimination, Yuhang Cao et al., NeurIPS 2021.

**Questions:**

I can understand that the prototype of a novel class is biased due to the lack of samples, and it can be easily misclassified as a base class. Intuitively, if the novel class is prone to be classified as the most similar base class, interpolation among such features could exacerbate this issue. But why is the use of weighted base prototypes to calibrate novel ones beneficial in enhancing the discriminability of such a prototype while reducing the discriminability of base prototypes (lines 217-218, 238-239)?. And could you please clarify the definition of discriminability?

**Limitations:**

As mentioned in the weaknesses section, the proposed idea is not novel at all. It unfortunately is just a tiny incremental contribution which is rather insufficient for a publication in top tier conference such as NeurIPS.

---

> ### Author Rebuttal · Authors · 2023-08-09
>
> **Q1** The issue of misclassification has already been observed and studied in previous few-shot learning works. Therefore, the poor performance of new classes is not surprising in the task of Few-Shot Class-Incremental Learning (FSCIL).
>
> **A1** We thank the reviewer for the suggestion.  **We must emphasize that our paper not only illustrates the phenomenon of low performance on new classes but also investigates the underlying causes behind this phenomenon**. Furthermore, based on our analysis of this phenomenon, we propose our  approach (i.e., TEEN). These analyses are unique in the FSCIL field and we believe that these analysis can provide new insights for the FSCIL field. Besides, if there are some papers that closely resemble our experimental analysis and methods, we would appreciate it if you could directly point out those relevant papers. We will include a detailed discussion of these papers in the final version of our paper.
>
> ---
>
> **Q2** Forming connections across samples from base and novel classes is not a new concept. For instance, FADI[1] discovered that a novel class may implicitly leverage the knowledge of multiple base classes to construct its feature space. It then builds a discriminative feature space for each novel class via association and discrimination steps. [1] Few-Shot Object Detection via Association and Discrimination, Yuhang Cao et al., NeurIPS 2021.
>
> **A2**  Thank you for providing this helpful citation. We differentiate our method from [1] based on the following perspectives:
>
> - **Setting and Motivation**: [1] focus on the Few-Shot Object Detection (**FSOD**) task and argue that directly fine-tune the model pre-trained on all base classes with abundant samples will lead a inferior performance of new classes. However, it donot analyze the advantages of connecting new and base classes in the FSOD scenario. In other words, the motivation behind connecting base and novel classes in the FSOD scenario is unclea, and there is a lack of experimental insights in [1]. In contrast, our paper focuses on the **FSCIL** task and first explore the reasons for the low performance of new classes in FSCIL. We find that new classes are generally easily misclassified into the most similar base classes, and we also empirically find that feature extractors are still able to characterize the semantic similarity between new and old classes even if they have not been trained on new classes. Based on these unique analyses, we achieve the goal of calibrating the new class prototype with the old class by utilizing feature extractors that have only been trained on the base class and its implied semantic similarity
> - **Methodology**: [1] adopt a two-stage (i.e., association and discrimination) learning method to learn a more discriminative novel classiﬁer. However, it adopt the  WordNet [2] as an auxiliary to describe the semantic similarity of each classes and use the Lin Similarity. Besides, [1] align the novel classes to the most similar base class. After that, [1] adopt a *Set-Specialized Margin Loss* to explicitly improve the discriminability. In contrast,  our method TEEN propose to leverage the feature extractor trained *only on base classes* to characterize the semantic similarity between the base and novel classes and donot extra semantic characterization model, which is more simple. Besides, TEEN directly used the cosine similarity to calibrate the new prototypes based on the weighted base prototypes and donot involve any training module or stage, which is more efficient.
>
> [2] George A Miller. Wordnet: a lexical database for english.
>
> ---
>
> **Q3** why is the use of weighted base prototypes to calibrate novel ones beneficial in enhancing the discriminability of such a prototype while reducing the discriminability of base prototypes (lines 217-218, 238-239)?. And could you please clarify the definition of discriminability?
>
> **A3**  Thanks for your question. We will address your questions separately in two ways.
>
> -  **Firstly**, we clarify the definition of discriminability in the FSCIL context. Here, when we talk about *discriminability*, we mainly refer to **prototypes**, which are the class centers of each class. From the results, if the classification accuracy of a class improves, we can infer that the discriminability of that category's prototype has also improved.
> - **Secondly**, we need to emphasize that in FSCIL, the model is required to recognize both new and base classes *simultaneously*. In our approach, we utilize semantic similarity to calibrate the new class prototypes based on a weighted combination of prototypes from semantically related base classes. Intuitively, this process brings the new class prototypes closer to the base class prototypes to a certain extent. Consequently, some samples from the base classes may be mistakenly classified as the calibrated new classes. We comprehensively analyze the potential negative (in Figure 3 and section 4.2) and positive effect of our calibration method (section 5), which show the efficiency of TEEN. Please refer to section 5.3 for the effectiness of semantic-based weight.
>
> ---
>
> **Q4** The proposed idea is not novel at all.
>
> **A4** We recapitulate our contributions. First, we empirically find that new class performance was
> much lower than the base class in previous methods. Then, we explore the causes of this phenomenon. These analyses are first proposed in the FSCIL field, which points out that we should pay more attention to the performance of new classes. Besides, we observe the feature extractor trained on base classes can also depict the semantic similarity between the base and new classes. Based on these analyses, we propose TEEN, which not only achieved a higher average accuracy but also improved the accuracy of new classes (**10.02% ∼ 18.40% better than the runner-up method**). Finally, we also validate TEEN on the Few-Shot Learning (FSL) task, which can also show competitive performance.

---

> > ### Comment · Reviewer_aEVg · 2023-08-17
> >
> > 1. I thought that the reasons behind misclassification from base to novel classes are two-fold: 1) the standard training method for deep neural networks typically involves empirical risk minimization (ERM) [V. Vapnik. Principles of risk minimization for learning theory. In NeurIPS], and 2) misclassifying novel classes has much less risk compared to base classes because the number of samples belonging to base classes is much larger than the number of samples belonging to novel classes. Besides, it is intuitive that the classifier misclassifies the new classes to their corresponding most similar base classes. So, that is why FADI used an association step to build the bridge between base and novel categories based on the most similar semantic information between them.
> > 2. I disagree with the statement that  'FADI does not analyze the advantages of connecting new and base classes in the FSOD scenario. In other words, the motivation behind connecting base and novel classes in the FSOD scenario is unclea'.  Actually, this paper provided the benefits of associating base and novel classes via pseudo-dual labels, and it also explained the motivation for this process.
> > 3. Based on your conclusion that new classes are prone to their most similar base classes, why using the most similar prototype of the base class reduces the performance on novel classes, as shown in Figure 5d.

---

> > > ### Author Response · Authors · 2023-08-19
> > > **Further Clarification of Reviewers' Concerns: Providing Deeper Insights**
> > >
> > > Thank you for your response. Before addressing your concerns individually, we would like to re-emphasize that [1] and TEEN are focused on different tasks, namely FSOD and FSCIL, respectively. Furthermore, let us recapitulate the main differences between [1] and TEEN.
> > >
> > > - First, [1] heavily relies on an external semantic similarity source, such as WordNet, to identify the most similar base class for each novel class. As mentioned in [1], the assigning policy is a crucial component in the association step. Notably, [1] associates **only one** most similar base class with each corresponding new class. In contrast, TEEN substantiates the overlooked characteristics (i.e., the feature extractor only trained on base classes can also characterize the semantic similarity between the base classes and unseen novel classes) of the feature extractor through quantitative and qualitative analysis, thus breaking free from the reliance on additional similarity characterization tools. Furthermore, TEEN utilizes the semantic similarity captured by the feature extractor to weight the prototypes of the base classes. It then calibrates the prototypes of the new classes based on the weighted semantic information from the base classes.
> > > - Secondly, in the FSOD task, the method proposed in [1] leverages the reuse of base class samples for balanced fine-tuning, leading to an alignment of the feature distribution. In contrast, TEEN does not utilize any base class samples or employ additional training modules when performing incremental recognition of the novel classes.
> > >
> > > **A1**: First and foremost, it is essential to emphasize that in the current FSCIL task, fixing the feature extractor is a highly common practice. Furthermore, not all methods involve complex ERM (Empirical Risk Minimization) training during incremental learning. As stated in section 2.2, these methods typically adopt a frozen feature extractor and utilize the prototypes for new classes to achieve the recognition of novel classes. Therefore, the low performance of the novel classes in these methods may not be directly explained by the training process, where misclassifying new classes as old classes could easily degrade ERM. Besides, *we agree that it may not be intuitive for the novel classes to be misclassified into the most similar base classes*. The low performance of the novel classes could also be attributed to inaccurate representations of the individual novel classes, leading to misclassifications among the novel classes themselves. We supported our conclusions through quantitative analysis based on empirical results in Section 3 rather than relying solely on subjective intuition. We believe these experimental analyses are meaningful for understanding tasks related to few-shot scenarios.
> > >
> > > **A2**: We apologize for your confusion. Our work delves into the advantages of connecting novel and base classes from an empirical standpoint, which is not thoroughly investigated in FSDI. In Section 3, we conducted comprehensive experiments to quantitatively explore and draw conclusions based on our observations, which motivated our method. In contrast, the motivation provided in FSDI primarily relies on qualitative reasoning. We believe that our experimental analysis is insightful and valuable.
> > >
> > > **A3**: In Figure 5d, we demonstrate that the utilization of semantic-based similarity (as shown in Eq5 and Eq6) and directly aligning new classes to their respective most similar $K$ base classes (as described in SimpleTEEN in Section 5.3) does not effectively improve the performance of the new classes compared to TEEN. As previously mentioned in the comment, we conducted a detailed analysis of the misclassifications of new classes. We collected statistics on how novel classes were misclassified into **the top 10 most similar base classes** (i.e., Table 2) rather than just one base class. This finding indicates that while the feature extractor trained solely on base classes can capture semantic similarity to some extent, it may not accurately identify the single most similar base class. Therefore, we propose calibrating the novel prototypes by the *weighted* base prototypes with semantic-based similarity. We appreciate your constructive feedback and will provide a more detailed discussion in the final version.
> > >
> > > We hope that our response has addressed your concerns and alleviated any doubts you may have had. If you have any further questions or require additional clarification, we are more than happy to engage in further discussion.

---

> > > > ### Comment · Reviewer_aEVg · 2023-08-20
> > > >
> > > > Thank you for the authors' response. It has mostly addressed my concerns. I would like to increase the score to 5. However, I still think that the novelty of the work is limited and incremental, despite its efficiency.

---

> > > > > ### Author Response · Authors · 2023-08-21
> > > > > **Appreciation for Reviewer aEVg**
> > > > >
> > > > > Thank you for adjusting your score. We are delighted that we were able to address your concern. Regarding the novelty and contribution of our paper, we have already elaborated in the global comment from the point of view of *experimental observations, simplicity of the methodology, and compatibility with the FSL task*. Besides, thank you for your constructive comments, we will provide a more in-depth discussion on this in the final version.

---

### Official Review · Reviewer_wHh2 · 2023-07-06

**Soundness:** 3 good
**Presentation:** 2 fair
**Contribution:** 2 fair
**Rating:** 5
**Confidence:** 4

**Summary:**

This paper presents a novel training-free calibration approach for few-shot class incremental learning. The authors make an observation that one main problem with FSCIL is that data of new classes can be easily mis-classified as base session classes. By utilizing the well-calibrated embeddings of base session classes to help embeedings of the new classes, the author improve the performance of FSCIL.

**Strengths:**

1. The observation of new classes being confused as old classes is somewhat novel.
2. The proposed classifier-calibration method is interesting and novel.
3. The proposed method is simple yet effective.
4. Detailed ablation studies have been performed on the introduced hyper-parameters.

**Weaknesses:**

1. One of the reviewer's concern is in terms of the robustness of the method to hyper-parameter \alpha and \tau. Are the optimal hyper-parameters the same for different datasets or different incrementing procedures (i.e., how many classes per incremental session)?
2. In addition, the author should provide more detailed justification for the design of the method. Why use this simple linear interpolation for calibration? Are there any empirical observations or theory to support this design?

**Questions:**

1. I suggest the author provide more ablation study on robustness of hyper-parameter on more dataset.
2. I suggest the author provide more justification on the designing of the method (i.e., why this simple linear interpolation form). Such justification can be empirical visualizations, theoretical analysis, etc.

**Limitations:**

The authors have adequately addressed the limitations.

---

> ### Author Rebuttal · Authors · 2023-08-10
>
> **Q1** One of the reviewer's concern is in terms of the robustness of the method to hyper-parameter $\alpha$ and $\tau$. Are the optimal hyper-parameters the same for different datasets or different incrementing procedures (i.e., how many classes per incremental session)?
>
> **A1** We are sorry for your confusion. In fact, we clearly show the the value of $\alpha$ and $\tau$ on different benchmarks in line 266. The optimal value of $\tau$ is stable for all benchmark datasets and the $\alpha$ is different. A more detailed ablation studies on the $\alpha$ and $\tau$  are shown in Figure 5. Regarding the number of classes per incremental session, we introduce the base experiment setting in lines 254 to 256 and provide detailed settings in Section 4.1 of the supplementary material. Please refer to the corresponding section mentioned above for the relevant experimental settings.
>
> ---
>
> **Q2** In addition, the author should provide more detailed justification for the design of the method. Why use this simple linear interpolation for calibration? Are there any empirical observations or theory to support this design?
>
> **A2** We are sorry for your confusion. The format of linear interpolation is simple and widely-used in different machine learning or deep learning fields, such as mixup [1] and LVQ algorithm [2]. A simple and intuitive understanding is that the form of linear interpolation *pulls a vector closer towards another vector in terms of direction*. To visualize this more, we have used a toy example from 2d gaussian distribution for a simple demonstration. Please refer to **the Figure 1 in the Rebuttal PDF** for a more detailed introduction. In addition to this empirical interpretation, the empirical observation in Section 3 also strongly supports our design. Specifically, we empirically find the reason for the low performance of the new class is that it is heavily misclassified into the most similar base classes. Therefore, we can improve the performance of the new class by pulling the biased new prototypes towards the prototypes of the most similar base class.
>
>
>
> [1] Zhang, Hongyi, et al. mixup: Beyond Empirical Risk Minimization. https://arxiv.org/abs/1710.09412
>
> [2] Learning vector quantization. https://en.wikipedia.org/wiki/Learning_vector_quantization
>
> ---
>
> **Q3** I suggest the author provide more ablation study on robustness of hyper-parameter on more dataset.
>
> **A3** We thank the reviewer for the suggestion. Please refer to **the Figure 2 in the Rebuttal PDF** for ablation studis of hyper-parameters (i.e., $\alpha$ and $\tau$) on more datasets. To better illustrate the trends, we have included the replicated trend figure5 (a)) in this rebuttal pdf.  We observed that the trend of the hyperparameter $\alpha$ remains consistent across all benchmark datasets. Although the trend of the hyperparameter $\tau$ may vary slightly between the CUB200, CIFAR100, and miniImageNet datasets, maintaining uniform settings still allows TEEN to maintain competitive performance. Notably, the trend of $\tau$ in miniImageNet and CIFAR100 datasets is perfectly identical. We hypothesize that the slight difference observed in CUB200 may be attributed to the fact that it is a *fine-grained* dataset.

---

> > ### Comment · Area_Chair_t9TB · 2023-08-16
> >
> > Dear reviewer wHh2,
> >
> > A detailed author rebuttal is in. Please share your thoughts post rebuttal.
> >
> > Thanks.
> >
> > AC

---

> ### Author Response · Authors · 2023-08-19
> **For requests for further discussion**
>
> We appreciate your constructive feedback and have provided corresponding responses. We are more than willing to address any further questions or concerns you may have. Please feel free to ask anything or discuss any further points of clarification.

---

> > ### Comment · Reviewer_wHh2 · 2023-08-20
> >
> > Thanks for the clarification!
> > I think the author's response have mostly addressed my concerns.
> > Therefore, I would like to increase the score to 5.

---

> > > ### Author Response · Authors · 2023-08-21
> > > **Appreciation for Reviewer WHh2**
> > >
> > > Thank you for adjusting your score. We are happy we managed to address your concern.

---

### Official Review · Reviewer_RFbX · 2023-07-06

**Soundness:** 3 good
**Presentation:** 4 excellent
**Contribution:** 3 good
**Rating:** 5
**Confidence:** 5

**Summary:**

The paper closely looks at problems in prototypical networks and methods in the context of few shot class incremental learning. The authors observe semantic similarity between new and base prototypes along with new classes being misclassified regularly as semantically similar base classes. Towards this they use a training free calibration strategy to guide the new ill calibrated prototypes using the base prototypes. Essentially bridging the lack of training data and suppressing bias.

**Strengths:**

- The major strength of the paper lies in the simplicity of the method. Instead of compromising the base class performance they approach the problem from a new class orientation. Doing so relieves the method from the convoluted approaches of feature space reservation[1], meta learning schemes[2] or self supervision[3]. Instead the methods strength lies in utilising the uncompromised power of the base prototypes.
- The fact that almost all evaluation metrics one way or the other focus on the novel class performance and not the less interpretable average accuracy and performance decay rate makes the paper even more attractive. This shows commitment towards pushing the frontier in aspects previously underexplored and facilitating a fair and thorough comparison.
- The work is constructed clearly and simply. It seems obvious that the authors followed a train of thought. They end up convincing the reader on why each module is essential. It generally reads very smoothly, hopping from hypothesis to observations to results and repeats. For example section 3 which tries to “Understand the reason for poor performance in new classes” leads smoothly into the observations for that section which motivates the section on calibration (section 4) and this section takes a similar hypothesis-observation-result route.
- The fact that almost all evaluation metrics one way or the other focus on the novel class performance and not the less interpretable average accuracy and performance decay rate makes the paper even more attractive. This shows commitment towards pushing the frontier in aspects previously underexplored and facilitating a fair and thorough comparison.


**Weaknesses:**

- Session 4 uses a variety of terminology. Using a consistent term for each prototype could potentially read better. (For example ill-calibrated new prototypes, new prototypes, biased prototypes)
- In relation to prior work it seems only Section 1 is truly dedicated to the literature. And even then it seems the prior works that this paper branches out from are not discussed in appropriate detail. Prior methods from the 3 other papers from Table 1 and Table 2 are not detailed in any preceding or subsequent section. Their would be reason to do so as the method directly criticises (line 86-87 or line 190-191) prior works. Prefacing in better detail how previous methods mitigate biases could be essential in understanding the state of the literature and motivate TEEN.
- [4] show in “Simpleshot” that L2 normalisation of prototypes improves performance. This is also confirmed by [5] in their nearest neighbour method for few shot learning. The authors overlooked this crucial feature transformation which is commonly conducted in previous studies on Non Parametric methods for FSCIL for example the NoNPC method by Oh et al. [6] This omission somewhat limits the comprehensiveness of their findings despite the improved results given the prevalnece of the method.
- [7], [8] are missing comparisons. Therefore, the tables that compare the method to the current sota would should less prominent results.

Minor Remarks and Typos:
- 213: “rely” should be “relies”
- 2: “scarce”
- The use of tau parameter in equation (5) might be better understood if shifted to equation 6 as S_{b,n} is a scalar and in any case it seems more common that a temperature scaling parameter appears in the softmax. Would recommend to make that alteration.
- 73: “neglect” should be “negligence”


[1] Zhou, Da-Wei, et al. "Forward compatible few-shot class- incremental learning." Proceedings of the IEEE/CVF conference on computer vision and pattern recognition. 2022.

[2] Zhang, Chi, et al. "Few-shot incremental learning with continually evolved classifiers." Proceedings of the IEEE/CVF conference on computer vision and pattern recognition. 2021.

[3] Ahmad, Touqeer, et al. "Few-shot class incremental learning leveraging self-supervised features." Proceedings of the IEEE/CVF Conference on Computer Vision and Pattern Recognition. 2022.

[4] Yan Wang, Wei-Lun Chao, Kilian Q Weinberger, and Laurens van der Maaten. Simpleshot: Revisiting nearest-neighbor classification for few- shot learning. arXiv preprint arXiv:1911.04623, 2019.

[5] Wang, Guangpeng, and Yongxiong Wang. "Self-attention network for few-shot learning based on nearest-neighbor algorithm." Machine Vision and Applications 34.2 (2023): 28.

[6] Oh, J., & Yun, S.-Y. (2022). Demystifying the Base and Novel Performances for Few-shot Class-incremental Learning. http://arxiv.org /abs/2206.10596

[7]  Peng, Can et al. Few-Shot Class-Incremental Learning from an Open-Set Perspective, ECCV 2022

[8] Yibo Yang, Haobo Yuan, Xiangtai Li, Zhouchen Lin, Philip Torr, and Dacheng Tao. Neural collapse inspired feature-classifier alignment for few-shot class-incremental learning. In International Conference on Learning Representations 2023

**Questions:**

- See weaknesses.
- Discussions and comparisons to the most recent sota are necessary


**Limitations:**

The authors have adequately mentioned the limitations. And motivate the setting where the pretrain base classes being from different data distribution as the incremental new classes is actually a more realistic setting.

---

> ### Author Rebuttal · Authors · 2023-08-10
>
> **Q1** Session 4 uses a variety of terminology. Using a consistent term for each prototype could potentially read better. (For example ill-calibrated new prototypes, new prototypes, biased prototypes)
>
> **A1** We thank the reviewer RFbX for the kind and meaningful writing suggestions. We will fix the terminology and typos in the final version.
>
> ---
>
> **Q2** In relation to prior work it seems only Section 1 is truly dedicated to the literature. And even then it seems the prior works that this paper branches out from are not discussed in appropriate detail. Prior methods from the 3 other papers from Table 1 and Table 2 are not detailed in any preceding or subsequent section. Their would be reason to do so as the method directly criticises (line 86-87 or line 190-191) prior works. Prefacing in better detail how previous methods mitigate biases could be essential in understanding the state of the literature and motivate TEEN.
>
> **A2** We thank the reviewer RFbX's constructive suggestions. We summarize the ProtoNet, CEC and FACT in the response and a more detailed introduction will be included in the final version. These three baselines freeze the feature extractor to mitigate the forgetting problem in incremental learning scenarios and adopt the prototype-based classifier to mitigate the overfitting problem in the few-shot scenario. We will summarize their differences separately.
>
> - The ProtoNet is a popular baseline method in the FSCIL field. It only pre-trains the feature extractor on base classes with a vanilla cross-entropy loss and plugs the prototype of novel classes to jointly recognize the base and novel classes.
> - The CEC follows a meta-learning paradigm and construct some pseudo few-shot task in the pre-training stage. Based on these pseudo few-shot tasks, CEC trains a classiﬁer adaptation module to adapt new classes in the incremental sessions.
> -  The FACT follows a forward-compatible paradigm to preserve some feature space for the incoming novel classes.
>
> **Summary**: Although these FSCIL methods achieve a competitive average performance of base and new classes, we observe that these methods still perform poorly on new classes (See section 3). Our method TEEN aims to *explicitly* mitigate the problem of poor performance on new classes and bridge the gap between base and novel classes and finally achieve better average performance.
>
> ---
>
> **Q3** [4] show in “Simpleshot” that L2 normalisation of prototypes improves performance. This is also confirmed by [5] in their nearest neighbour method for few shot learning. The authors overlooked this crucial feature transformation which is commonly conducted in previous studies on Non Parametric methods for FSCIL for example the NoNPC method by Oh et al. [6]
>
> **A3** We thank the reviewer RFbX for providing the related works. We agree that the widely-used $L_2$ normalization of the prototypes is beneficial in few-shot related tasks. Therefore, we do not overlook yet adopt this normalization in Line219-Line220 (Eq6). Besides, although [6] also adopts the $L_2$ normalization. **There still exists main differences between our paper and NoNPC [6]**. For example, we explicitly *observe and explore the reason for the low performance of new classes* in the FSCIL scenario, which is absent in [6]. Based on our **unique** empirical analysis, we propose *explicitly* calibrating the novel prototypes through the implicit semantic similarity overlooked by existing works with a *training-free* method. A more detailed discussion with NoNPC will be included in the final version.
>
> ---
>
> **Q4** [7], [8] are missing comparisons. Therefore, the tables that compare the method to the current sota would should less prominent results.
>
> **A4** We appreciate the valuable suggestions from the reviewers regarding the strong baseline methods. We acknowledge that TEEN may not directly outperform the missing baseline methods [8]. However, in our supplementary material, we demonstrate the potential of TEEN as a *plug-and-play* module in Figure 1 and Figure 2. We have tried to integrate TEEN with the mentioned method NC-FSCIL and fine-tune TEEN's two hyper-parameters in the evaluation stage of NC-FSCIL [8]. The plug-and-play nature of TEEN also allows it to achieve competitive performance.
>
> |                  |   0   |   1   |   2   |   3   |   4   |   5   |   6   |   7   |   8   |
> | :--------------: | :---: | :---: | :---: | :---: | :---: | :---: | :---: | :---: | :---: |
> |    ALICE [7]     | 80.60 | 70.60 | 67.40 | 64.50 | 62.50 | 60.00 | 57.80 | 56.80 | 55.70 |
> |   NC-FSCIL [8]   | 84.02 | 76.80 | 72.00 | 67.83 | 66.35 | 64.04 | 61.46 | 59.54 | 58.31 |
> | NC-FSCIL w/ TEEN | 84.06 | 76.82 | 72.12 | 67.86 | 66.41 | 64.21 | 61.63 | 59.55 | 58.43 |

---

> > ### Comment · Reviewer_RFbX · 2023-08-19
> >
> > Thank you for the authors' response. After reading the rebuttal and the comments of the other reviewers, my concerns have been thoroughly addressed and thus I maintain towards the acceptance of this work.

---

### Author Rebuttal · Authors · 2023-08-10

We express our profound gratitude to the reviewers for their insightful and valuable comments. We are pleased that the reviewers find the simplicity and efficiency of the proposed method TEEN (RFbX, wHh2, 3Zyg, TNew) and our work clear and easy to follow (RFbX, 3Zyg, aEVg).  They also consider our method and empirical analysis interesting, novel and well-motivated (wHh2, aEVg, 3Zyg).  The reviewer wHh2 acknowledges our detailed ablation studies. In particular, reviewer RFbX acknowledges that our approach **takes a new class orientation, pushing the frontier in previously underexplored aspects and facilitating a fair and comprehensive comparison** and praises our work as **attractive**.

Before our rebuttal, we recapitulate our *contribution, novelty and some easily overlooked shining points*:

- We empirically find that new class performance was much lower than base class in previous methods. Then, we explore the causes of this phenomenon. *These analyses are first proposed in the FSCIL field, which points out that we should pay more attention to the performance of new classes*.

- We also observe that the feature extractor, solely trained on base classes, can capture the semantic similarity between the base and new classes.  Based on these analyses, we propose TEEN, which not only achieved a higher average accuracy but also improved the accuracy of new classes.

- In addition to the FSCIL task, we also validate TEEN on the Few-Shot Learning (FSL) task, which can also show competitive performance. In contrast, existing methods lack effectiveness for **both** FSCIL and FSL tasks.



To better address the reviewer's concerns, we conducted experiments primarily focusing on three aspects: cross-dataset evaluation, hyper-parameter robustness and the intuitive explanation of TEEN.

- **Cross-dataset few-shot class-incremental learning scenario**.  We *reuse* the feature extractor pretrained on the base session of **CUB200** and use this feature extractor for the incremental learning on **miniImageNet** datasets. Specifically, Session 0 consists of 100 classes from the CUB200 dataset, while the incremental sessions (Session 1-6) include classes from the miniImageNet dataset in a 5-way 5-shot format. In the cross-dataset incremental learning scenario, the performance of CEC and FACT  show a significant drop, while **the TEEN continues to perform competitively**. We speculate that these methods (CEC and FACT) may have designed overly complex modules (e.g., classiﬁer adaptation module in CEC) or mechanisms (e.g., the forward-compatible paradigm in FACT), which led to excessive adaptation to the pre-training dataset. In contrast, TEEN relies solely on the underlying semantic similarity relationships to calibrate the new class prototypes, which may make TEEN more resilient to dataset shifts.

|      | 0 (CUB200) |   1   |   2   |   3   |   4   |   5   |   6   |
| :--: | :--------: | :---: | :---: | :---: | :---: | :---: | :---: |
| CEC  |   75.85    | 65.34 | 63.21 | 61.30 | 59.24 | 57.29 | 55.21 |
| FACT |   75.90    | 63.02 | 60.12 | 58.42 | 55.32 | 53.54 | 51.48 |
| TEEN |   77.26    | 68.42 | 66.32 | 65.19 | 62.14 | 59.43 | 58.56 |

- **The robustness of the hyper-parameters $\alpha$ and $\tau$**. We show the performance trend with respect to $\alpha$ and $\tau$ on each benchmark dataset **in the Rebuttal PDF (Figure 2)**. To better illustrate the trends, we have included the replicated trend figure 5 (a)) in this rebuttal pdf.  We observed that the trend of the hyperparameter $\alpha$ remains consistent across all benchmark datasets. Although the trend of the hyperparameter $\tau$ may vary slightly between the CUB200, CIFAR100, and miniImageNet datasets, maintaining uniform settings still allows TEEN to maintain competitive performance. Notably, the trend of $\tau$ in miniImageNet and CIFAR100 datasets is perfectly identical. We hypothesize that the slight difference observed in CUB200 may be attributed to the fact that it is a *fine-grained* dataset.
- **An example that provides an intuitive explanation of TEEN using a toy example**. To better help reviewers understand the design of our calibration method, we show an intuitive visualization using a toy dataset sampled from a 2D Gaussian distribution (**Figure 1 in Rebuttal PDF**). Specifically, we constructed four classes, each corresponding to a Gaussian distribution. Three of these classes served as the base classes (with *a large number of* samples), while one class represented the new class (with *a small number of* samples). By applying TEEN to the new class, we can visually observe that the calibrated prototypes of the new class are closer to the desired class prototypes than their initial uncalibrated counterparts.

**We have included the requested comparison and discussions in the corresponding answers and the rebuttal PDF**. The relevant papers mentioned in the response or any further detailed discussions will be included in the final version. Please check the answers to specific comments and Rebuttal PDF for details.

---

### Decision · Program_Chairs · 2023-09-21

**Decision:**

Accept (poster)

**Comment:**

This paper presents an approach for class-incremental learning in a few-shot framework. The core of the approach relies on similarity among classes. While I agree with some of the reviewer comments that this is not the most novel paper in terms of ideas: using similarity among classes for the incremental learning problem, it takes this simple idea and executes it well. It also has detailed experimental evaluations to support the proposed approach.

The rebuttal and the discussions that ensued have also convinced a couple of reviewers to increase their scores, while the others maintained their positive evaluations. Overall, the paper clearly shows something that works and how it works. Based on this, the AC recommends the paper for acceptance.